# Tight Clusters Make Specialized Experts

**Stefan K. Nielsen**[*]
FPT Software AI Center
stefannvkp@fpt.com

**Rachel S.Y. Teo**[*]
Department of Mathematics
National University of Singapore
rachel.tsy@u.nus.edu

**Laziz U. Abdullaev**
Department of Mathematics
National University of Singapore
laziz.abdullaev@u.nus.edu

**Tan M. Nguyen**
Department of Mathematics
National University of Singapore
tanmn@nus.edu.sg

## Abstract

Sparse Mixture-of-Experts (MoE) architectures have emerged as a promising approach to decoupling model capacity from computational cost. At the core of the MoE model is the router, which learns the underlying clustering structure of the input distribution in order to send input tokens to appropriate experts. However, latent clusters may be unidentifiable in high dimension, which causes slow convergence, susceptibility to data contamination, and overall degraded representations as the router is unable to perform appropriate token-expert matching. We examine the router through the lens of clustering optimization and derive optimal feature weights that maximally identify the latent clusters. We use these weights to compute the token-expert routing assignments in an adaptively transformed space that promotes well-separated clusters, which helps identify the best-matched expert for each token. In particular, for each expert cluster, we compute a set of weights that scales features according to whether that expert clusters tightly along that feature. We term this novel router the Adaptive Clustering (AC) router. Our AC router enables the MoE model to obtain three connected benefits: 1) faster convergence, 2) better robustness to data corruption, and 3) overall performance improvement, as experts are specialized in semantically distinct regions of the input space. We empirically demonstrate the advantages of our AC router over baseline routing methods when applied on a variety of MoE backbones for language modeling and image recognition tasks in both clean and corrupted settings. The code is publicly available at https://github.com/stefvk/ACMoE.

## 1 Introduction

Scaling up model capacity continues to deliver substantial performance gains across a wide range of tasks, with particularly impressive results in visual representation learning and language modeling (Alexey, 2020; Bao et al., 2021; Radford et al., 2019; Raffel et al., 2020; Nguyen et al., 2023). However, larger models incur growing computational costs, prompting increasing research into Sparse Mixture-of-Experts models (MoE), which offers a promising avenue to balancing model scale with efficiency by activating only sub-modules, termed *experts*, of the network during training and inference (Shazeer et al., 2017; Fedus et al., 2022; Lepikhin et al., 2020; Nguyen et al., 2025). This approach has been shown to achieve better performance than dense models with nearly constant computational overhead on tasks from speech recognition, image recognition, machine translation, and language modeling (Riquelme et al., 2021; Kumatani et al., 2021; Lepikhin et al., 2020; Teo & Nguyen, 2025a).

At the core of the MoE layer is the learned router which assigns inputs to the relevant experts. The router must learn to segment the input space appropriately such that inputs and experts are well matched, enabling the experts to be trained on semantically similar data. This expert specialization

---

[*] Co-first authors. Please correspond to: stefannvkp@fpt.com and tanmn@nus.edu.sg

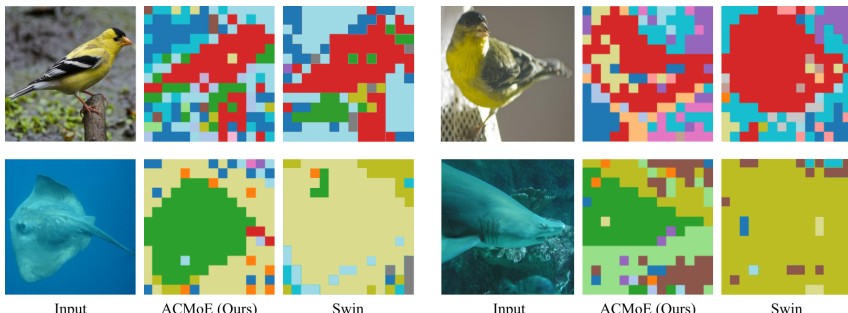

Figure 1: ACMoE discovers semantically distinct regions. We show 14x14 image reconstructions where patches are colored by assigned experts. **Top row:** Swin assigns large chunks of foreground and background to one expert (red), while ACMoE accurately discovers the bird and relevant foreground. **Bottom row:** When the background and foreground are hard to distinguish, Swin's router fails to register the stingray (left) or shark (right) and allocates one expert for virtually the entire image. ACMoE, however, discovers the semantically distinct regions, using one expert (green) to specialize on the foreground and different experts for the background.

allows MoE models to produce better representations than their dense counterparts while activating only a fraction of the total parameters. Recently, various methods have been proposed to find optimal expert-token matches, including linear programs (Lewis et al., 2021), cosine similarity-based rules (Chi et al., 2022), soft assignments via convex combinations of inputs (Puigcerver et al., 2023), and both top-k experts per token (Shazeer et al., 2017) and top-k tokens per expert (Zhou et al., 2022b). We note that the above approaches fundamentally rely on dot-products between inputs and experts to learn the corresponding assignment, which might be suboptimal in cases where the semantic regions are not easily discoverable in the high-dimensional feature space. Typically, we expect that the true underlying clusters present in the data will cluster on different, potentially disjoint, subsets of features, and may not be discoverable when using the full feature set. This phenomenon can lead to slow convergence as the experts are unable to specialize on semantically similar regions of the data, poor robustness as data contamination can spuriously assign inputs to unsuitable experts, and degraded overall downstream performance due to suboptimal input-expert matching.

**Contribution.** In this work, we propose the Adaptive Clustering (AC) router and corresponding Adaptive Clustering Mixture-of-Experts (ACMoE), a novel MoE method in which the router computes token-expert assignments in a transformed space that maximally identifies latent clusters in the data and more easily discovers the best-matched expert for each token. More specifically, we adaptively learn for each input which features best determine its cluster assignment and scale its features accordingly such that features that promote tight expert clusters are upweighted, and features that produce dispersed expert clusters are downweighted. This transformation accentuates the relevant characteristics of each input according to the specialization of the experts, thereby allowing the router to more easily discover the optimal input-expert allocation. Computing the routing assignments following this scheme produces three benefits: 1) *faster convergence* as experts are able to specialize more quickly by being allocated semantically similar inputs, 2) *better robustness* as latent clusters are better separated, thereby minimizing the risk that data corruption erroneously assigns tokens to unsuitable experts, and 3) *better overall representations and downstream performance* due to improved expert specialization. In order to discover the key features per token and their corresponding weights, we present a feature-weighted clustering optimization perspective on the MoE framework and demonstrate how the clustering solution obtains the required feature weights. We show how these weights can be integrated into the routing mechanism such that routing takes place in a cluster-adaptive transformed space. We theoretically prove that our proposed routing mechanism learns the latent clustering structure of the data faster than standard routing mechanisms and that our mechanism is more robust to data contamination. Furthermore, our proposed method involves no learnable parameters and can be computed highly efficiently. In summary, our contributions are three-fold:

1. We develop the novel Adaptive Clustering router, a routing method in MoE architectures that computes token-expert assignments in a transformed space that promotes separation of latent clusters in the data and more easily identifies the best-matched expert for each token.

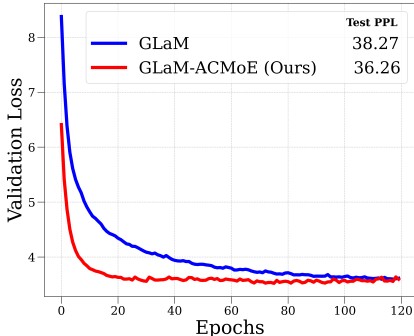 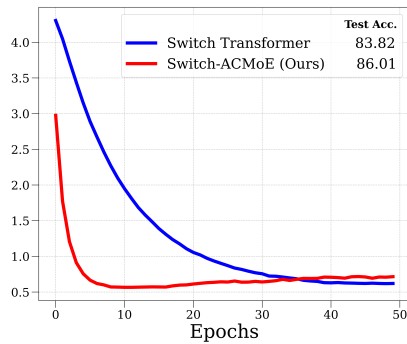

Figure 2: Fast Convergence of ACMoE. **Left:** Convergence speed on WikiText-103 pretraining using the Generalist Language Model (Du et al., 2022) backbone. **Right:** Convergence speed on Banking-77 finetuning using the Switch Transformer (Fedus et al., 2022) backbone. Across both backbones and tasks, we observe substantially faster convergence. We display final test perplexity (PPL) and accuracy (Acc.), showing better overall performance as well.

2. We propose a feature-weighted clustering optimization perspective on token-expert assignment and derive the optimal feature weights for adaptively transforming the input data for routing.

3. We derive a theoretical framework demonstrating how MoE robustness and convergence depend on the shape of latent clusters and the clustering geometry of the input space.

We empirically demonstrate that 1) the Adaptive Clustering router outperforms baseline routing methods in MoE architectures in large-scale tasks such as WikiText-103 language modeling and downstream finetuning, and ImageNet-1k object classification in both clean and contaminated settings, 2) the Adaptive Clustering router exhibits faster convergence than baseline methods, and 3) the Adaptive Clustering router attains these performance improvements for free – that is, with no learnable parameters and negligible computational overhead.

**Preliminaries.** We consider Transformer (Vaswani, 2017) based MoE architectures and follow the approach of previous work where the MoE layer is inserted after the self-attention layer within the Transformer, replacing the traditional feed-forward network (Fedus et al., 2022; Du et al., 2022; Liu et al., 2021). Let $\boldsymbol{x}$ be an input token with hidden representation $\boldsymbol{h} \in \mathbb{R}^d$ and $\boldsymbol{e}_1, \boldsymbol{e}_2, \ldots \boldsymbol{e}_N \in \mathbb{R}^d$ be the $N$ learnable expert embeddings for model hidden dimension $d$. The MoE layer selecting the top $k$ experts is described by the following equations:

$$\mathcal{K} := \mathrm{topk}_k(s_k) = \mathrm{topk}_k(\boldsymbol{h}^\top \boldsymbol{e}_k) \tag{1}$$

$$f^{SMoE}(\boldsymbol{h}) = \boldsymbol{h} + \sum_{k \in \mathcal{K}} g(\boldsymbol{h}^\top \boldsymbol{e}_k) f_k^{\mathrm{FFN}}(\boldsymbol{h}), \tag{2}$$

where $f_k^{\mathrm{FFN}}$ is the $k^{\mathrm{th}}$ expert feed-forward network, $s_k = \boldsymbol{h}^\top \boldsymbol{e}_k$ is the similarity score between token representation $\boldsymbol{h}$ and the $k^{\mathrm{th}}$ expert $\boldsymbol{e}_k$ and $g(\cdot)$ is a gating function often chosen as softmax, $g(s_k) = \exp(s_k)/\sum_{j \in \mathcal{K}} \exp(s_j)$. We refer to Eqn. 1 as the router, which learns the top $k$ best matched experts per token, and Eqn. 2 as the overall standard MoE layer.

**Organization.** We structure this paper as follows: In Section 2, we present a clustering optimization problem and show that its solution adaptively scales the feature space according to which dimensions promote tight clustering. In Section 3, we present how the solution to our clustering optimization problem can be built into our proposed AC router and we provide the full technical formulation of AC routing and Adaptive Clustering Mixture-of-Experts (ACMoE). We then present theoretical propositions on faster convergence and robustness. We empirically validate the advantages of AC-MoE in Section 4 and discuss related work in Section 5. We end with concluding remarks and future work in Section 6. Proofs, technical details, and further experiments are provided in the Appendix.

## 2 A CLUSTERING OPTIMIZATION PERSPECTIVE

We begin by examining the MoE router through the lens of feature-weighted clustering (Witten & Tibshirani, 2010; Friedman & Meulman, 2004; Brusco & Cradit, 2001; Gnanadesikan et al.,

1995). We explicitly model the router's task as learning a token assignment that groups together similar tokens. We consider the role of learnable feature weights in solving a clustering optimization problem to optimally reveal latent clusters and present an analytical solution for the optimal weights for any given routing assignment. We finally discuss how this solution improves the MoE router before providing the full formulation of our AC router and ACMoE in the next section.

## 2.1 CLUSTERING OPTIMIZATION

Let $\boldsymbol{h}_i = [h_{i1}, \ldots, h_{id}]^\top$ be the $i^{\text{th}}$ hidden representation and $D_{ij}$ denote the distance between $\boldsymbol{h}_i$ and $\boldsymbol{h}_j$. Given a distance metric $\rho_{ijq}$ between $h_{iq}$ and $h_{jq}$ over the $q^{\text{th}}$ dimension, the distance between $\boldsymbol{h}_i$ and $\boldsymbol{h}_j$ can be defined as $D_{ij}(\boldsymbol{w}) = \sum_{q \in [d]} w_q \rho_{ijq}$ for weights $\boldsymbol{w} = [w_1, \ldots, w_d]$ with $\sum_{q \in [d]} w_q = 1$ and $w_q \geq 0$ for all $q \in [d]$. The weights determine the global importance of the $q^{\text{th}}$ feature to the overall distance among representations.

Cluster analysis aims to divide the input set of $N$ objects into groups, where objects within the same group are more similar to each other than to those in other groups. This is formalized using a classifier $r(i) = k$, assigning the $i^{\text{th}}$ object to a group $k$. Then the optimal classifier $r^*$ minimizes a criterion $Q(r)$ that evaluates clustering quality:

$$r^* = \arg\min_r Q(r) = \sum_{k \in [E]} \frac{1}{N_k^2} \sum_{r(i)=k} \sum_{r(j)=k} D_{ij}(\boldsymbol{w}). \tag{3}$$

We expect that different groupings will cluster on different subsets of features. In particular, we wish to model the scenario that groupings exist in different latent subspaces with varying dependence on possibly disjoint subsets of features. We therefore replace the global feature weight $\boldsymbol{w}$ in Eqn. 3 with cluster-dependent feature weights, $\{\boldsymbol{w}_k\}_{k=1}^E$ for $E$ groups, which allows us to capture the differing feature dependencies of *each* cluster. Then, we can adapt the optimization problem with these cluster-dependent feature weights as follows:

$$(r^*, \{\boldsymbol{w}_k^*\}_{k=1}^E) = \arg\min_{r, \{\mathbf{w}_k\}} \sum_{k \in [E]} \frac{1}{N_k^2} \sum_{r(i)=k} \sum_{r(j)=k} D_{ij}^J(\boldsymbol{w}_k),$$
$$\text{such that} \sum_{q \in [d]} w_{qk} = 1, \quad \forall k \in [E], \tag{4}$$

where $D_{ij}^J(\boldsymbol{w}_k) = \sum_{l=1}^d w_{qk} \rho_{ijq} + \lambda J(\boldsymbol{w}_k)$ denotes the weighted distance between $i$ and $j$ combined with some regularization $J$ and regularization strength $\lambda$.

To avoid point-mass solutions in which we assign all weight to the single best-clustering feature, we set the regularizer to the Kullback-Leibler divergence between the feature weights $\boldsymbol{w}$ and the uniform distribution $\boldsymbol{u} = (1/d, \ldots, 1/d) \in \mathbb{R}^d$, denoted by $J(\boldsymbol{w}_k) = D_{\text{KL}}(\boldsymbol{u} \parallel \boldsymbol{w}_k)$. The regularization parameter $\lambda$ reflects our preference to maintain more or less features in the solution set.

## 2.2 MOE AS CLUSTERING OPTIMIZATION

Within the MoE framework with learnable routing, the router performs the role of the classifier $r : \mathbb{R}^d \to [E]$, which is learned via gradient descent to optimize the final output loss[1]. Therefore, we modify Eqn. 4 by fixing $r$ and focusing just on optimizing the criterion with respect to cluster-wise feature weights $\boldsymbol{w}_k$. Under this interpretation, the router learns via backpropagation to optimally allocate representations to experts, with representations adaptively transformed to maximally reveal the clustering structure of the input data. Eqn. 4 then becomes

$$\{\boldsymbol{w}_k^*\}_{k=1}^E = \arg\min_{\{\mathbf{w}_k\}} \sum_{k \in [E]} \frac{1}{N_k^2} \sum_{r(i)=k} \sum_{r(j)=k} D_{ij}^J(\boldsymbol{w}_k),$$
$$\text{such that} \sum_{q \in [d]} w_{qk} = 1, \quad \forall k \in [E]. \tag{5}$$

The following theorem presents the optimal weights per feature $q$ and cluster $k$:

---

[1]A top-$k$ router can straightforwardly be cast as the classifier in Eqn. 4 as $r : \mathbb{R}^d \to [E]^k$

**Theorem 1** (Optimal feature weights). *Let $s_{qk} := N_k^{-2} \sum_{r(i)=k} \sum_{r(j)=k} \rho_{ijq}$ be a measure of dispersion on the $q^{\text{th}}$ feature for the representations assigned to cluster $k$. Then, for a given router function $r : \mathbb{R}^d \to [E]$, the corresponding optimal weights $\{\boldsymbol{w}_k\}_{k \in [E]}$ that minimize the feature-weighted clustering optimization problem in Eqn. 5 are given by*

$$w_{qk} = \frac{\lambda/d}{s_{qk} + \alpha_k} \tag{6}$$

*for $(q, k) \in [d] \times [E]$, where $\{\alpha_k\}_{k \in [E]}$ are constants that for any $\lambda > 0$ satisfy*

$$\sum_{q \in [d]} \frac{1}{s_{qk} + \alpha_k} = \frac{d}{\lambda}. \tag{7}$$

The existence of $\alpha_k$ satisfying Eqn. 7 and the proof of Theorem 1 is provided in Appendix A.1. The optimal weights for a cluster $k$ given in Eqn. 6 take an intuitive form in that they are inversely proportional to the measure of dispersion in cluster $k$ along each dimension, $\boldsymbol{w}_k \propto \left[\frac{1}{s_{1k}}, \ldots, \frac{1}{s_{dk}}\right]$. Hence, the optimal cluster-wise feature weights scale features according to their contribution to forming tight clusters. Specifically, the solution weights upweight a feature $q$ if cluster $k$ clusters tightly (has small dispersion $s_{qk}$) along the feature $q$ and downweights a feature $p$ if cluster $k$ clusters loosely (has large dispersion $s_{pk}$) along feature $p$.

This method enables the MoE router to perform better token-expert matching. The cluster-wise feature weights $\boldsymbol{w}_k$ capture the features on which the $k^{\text{th}}$ expert is specialized, as large weights indicate those features are highly important to the identification of that expert cluster and small weights indicate those features are unimportant to identification of that expert cluster. Then, we can use $\boldsymbol{w}_k$ to scale the tokens to accentuate their features according to the specialization of the experts, thereby allowing the router to best identify the most suitable expert for each token. Note that this solution is local in that we learn the optimal weights adaptively *per cluster*, obtaining $\boldsymbol{w}_k$ for all $k \in [E]$, and so we compute a unique scaling of the feature space adaptively *per cluster* as well. Integrating these cluster-dependent weights which scale the feature space according to the identification of each expert into the MoE router obtains our AC routing method and corresponding ACMoE. We detail the AC router and ACMoE fully in the next section.

## 3  A Tight Cluster is a Specialized Expert

In this section, we demonstrate how we implement the solution weights from the clustering optimization problem in Eqn. 6 into the MoE routing mechanism, thereby obtaining the Adaptive Clustering router. We then provide the full technical formulation of our proposed routing method and corresponding ACMoE model. We also present theoretical results on how computing the routing assignments according to our framework promotes faster convergence and robustness.

### 3.1  Full Technical Formulation

We integrate the weights from Eqn. 6 into the Adaptive Clustering router transformation in Definition 1 which, for a cluster $k$, scales the dimensions of the feature space according to the $k^{\text{th}}$ expert's specialization on those features. Formally this is:

**Definition 1** (Adaptive Clustering Router Transformation $\boldsymbol{M}_k$). *Let $\mathcal{C}_k^\ell = \{\boldsymbol{h}_1^\ell, \ldots \boldsymbol{h}_{N_k}^\ell\}$ be the representations assigned to expert $k$ at layer $\ell$. Let $s_{qk}^\ell \in \mathbb{R}$ be a measure of a spread in the $q^{\text{th}}$ dimension for cluster $k$, such as mean absolute deviation $s_{qk}^\ell = \frac{1}{N_k} \sum_{i \in \mathcal{C}_k^\ell} |\boldsymbol{h}_{iq}^\ell - \bar{\boldsymbol{h}}_q^\ell|$. Then, the cluster-dependent router transformation for expert $k$ at layer $\ell$ is given by a diagonal matrix $\boldsymbol{M}_k^\ell := \text{diag}(1/s_{1k}^\ell, \ldots, 1/s_{dk}^\ell)$.*

We use the transformation $\boldsymbol{M}_k$ in Definition 1 to adaptively scale the feature space in which we perform token-expert matching. This obtains our Adaptive Clustering router and corresponding ACMoE layer, described in the following definition.

**Definition 2** (Adaptive Clustering Router and MoE Layer). *Let $\boldsymbol{h}^\ell \in \mathbb{R}^d$ be the hidden representation of an input, $\boldsymbol{e}_1^\ell, \ldots, \boldsymbol{e}_N^\ell \in \mathbb{R}^d$ be expert embeddings at layer $\ell$. Let $\boldsymbol{h}^{\ell-1} \in \mathcal{C}_{k^*}^{\ell-1}$ have been assigned*

*to expert $k^*$ in the previous layer. Let $M_{k^*}^{\ell-1} \in \mathbb{R}^{d \times d}$ be the Adaptive Clustering transformation (Definition 1) for input $h$ at layer $\ell - 1$. Let $g(\cdot)$ be the softmax function. Then the following equations describe the Adaptive Clustering router (Eqn. 8) and overall ACMoE layer (Eqn. 9):*

$$\mathcal{K} := \text{topk}_k(s_k) = \text{topk}_k(h^{\ell\top} M_{k^*}^{\ell-1} e_k^{\ell}) \tag{8}$$

$$f^{\text{ACMoE}}(h^{\ell}) = h^{\ell} + \sum_{k \in \mathcal{K}} g(h^{\ell\top} M_{k^*}^{\ell-1} e_k^{\ell}) f_k^{\text{FFN},\ell}(h^{\ell}). \tag{9}$$

**Remark 1.** *We see from Eqns. 8 and 9 that the standard MoE layer is recovered by setting the AC router transformation to the identity matrix, $M_k = I_d$ for all $k \in [E]$. Within our framework then, standard routing schemes implicitly assume all experts $k \in [E]$ depend equally on all dimensions.*

**Remark 2.** *The Adaptive Clustering router computes a dot-product between $h$ and experts $e_k$ with the dimensions scaled by the weights in $M_{k^*}$ and so is proportional to a Mahalanobis distance. Under this interpretation, we soft project the tokens and expert embeddings onto the axes of the feature space that best identify the expert cluster $k^*$.*

**Implementation details.** Given ACMoE requires the expert assignment from the previous layer to compute the routing assignment (Eqn. 8), ACMoE is only implementable after the first layer. Furthermore, we scale the measures of dispersion in $M_k^{\ell} = \text{diag}(1/s_{1k}^{\ell}, \ldots, 1/s_{dk}^{\ell})$ to have mean 1. This is to remove the effect of different clusters or features having different absolute magnitudes. Our method is concerned with identifying the key sets of features that contribute more or less to identification of the expert clusters, and so we wish to compute our scaling in a relative sense.

## 3.2 ADAPTIVE CLUSTERING PROMOTES ROBUSTNESS AND FAST CONVERGENCE

We now present theoretical propositions on the improved robustness and convergence speed of our method. The robustness of our method follows from better separation of expert-clusters. This produces a more stable assignment in which the probability of erroneously sending a token to unsuitable nearby experts decays exponentially with increased inter-cluster distance. Faster convergence follows from our AC routing method improving the conditioning on the Hessian of the loss with respect to the expert embeddings, enabling faster and more stable convergence of the router.

**Promoting robustness.** We begin with Lemma 1 stating that our AC transformation (Definition 1) increases the separation between clusters in the transformed space, followed by Lemma 2, which provides an explicit expression for the probability of incorrect expert assignment. To give the probability bound an exact form, we assume the cluster structure can be modeled as a Gaussian mixture model (GMM). We note that GMMs are a highly expressive and general framework, so this assumption does not place significant restrictions on our robustness analysis. We further assume that though clusters may overlap, they are well-separated along the features for which they cluster tightly[2].

**Lemma 1** (Adaptive Clustering Router Transformation Increases Cluster Separation). *Let the data be generated from a Gaussian mixture model with components, $g_c = \mathcal{N}(\mu_c, \Sigma_c)$ for $c \in [E]$. Without loss of generality, consider two expert clusters $c \in \{a, b\}$ where a token representation $h \sim g_a$ belongs to cluster $a$. Let $M_a = \text{diag}(1/s_{1a}, \ldots, 1/s_{da})$ be the router transformation constructed from the feature-wise dispersions, $s_{qa}$, of cluster $g_a$ for each feature $q \in [d]$ as given by Definition 1. Then the distance between cluster means in the $M_a$-transformed space, defined as $\|\mu_k - \mu_a\|_{M_a}^2 := (\mu_k - \mu_a)^\top M_a (\mu_k - \mu_a)$, is larger than in the original Euclidean space: $\|\mu_k - \mu_a\|_{M_a}^2 \geq \|\mu_k - \mu_a\|^2$.*

The proof is provided in Appendix A.2. In Lemma 2, we derive the probability of mis-assignment as a function of inter-cluster distance, showing how separation mitigates the effect of perturbations.

**Lemma 2** (Incorrect Assignment Probability). *Let $h \sim \mathcal{N}_{k^*}(\mu_{k^*}, \Sigma_{k^*})$ be a representation belonging to cluster $k^*$. Let $h' = h + \epsilon$ be contaminated by some 0-mean noise $\epsilon \sim (0, \Sigma_\epsilon)$. Let $k$ be the nearest, incorrect cluster to $k^*$. Let the inter-cluster mean distance between $k^*$ and $k$ be given by $\|\delta\mu\| := \|\mu_{k^*} - \mu_k\|$. Let the routing assignment be given by $r : \mathbb{R}^d \to [E]$ and denote the cumulative density of a standard normal distribution by $\Phi$. Then the probability of incorrect assignment is:*

$$\Pr(r(h') \neq k^*) = 1 - \Phi\left(\frac{\|\delta\mu\|^2}{2\sqrt{\delta\mu^\top(\Sigma_{k^*} + \Sigma_\epsilon)\delta\mu}}\right). \tag{10}$$

---

[2]Intuitively, this assumption captures the natural property that the semantic regions of the input space are distinct along the dimensions that best identify them.

**Remark 3.** *It is worth noting that since* $1 - \Phi(x) \sim (\sqrt{2\pi}x)^{-1}e^{-x^2/2}$ *for large* $x$ *and* $\sqrt{\delta\boldsymbol{\mu}^\top(\boldsymbol{\Sigma}_{k^*} + \boldsymbol{\Sigma}_\epsilon)\delta\boldsymbol{\mu}} = O(\|\boldsymbol{\mu}\|)$, *we find that the probability of incorrect cluster assignment as given by Eqn. 10,* $\Pr(r(\boldsymbol{h}') \neq k^*) = e^{-O(\|\delta\boldsymbol{\mu}\|^2)}$ *is an exponentially decreasing function in* $\|\delta\boldsymbol{\mu}\|$.

The proof is provided in Appendix A.2. Combining Lemmas 1 and 2, we directly obtain that the probability of erroneous assignment using the AC router is exponentially smaller than under a standard routing scheme. This is formalized in Proposition 1, given by:

**Proposition 1** (Robustness of ACMoE). *Consider an expert assignment setting for the representation* $\boldsymbol{h} \sim \mathcal{N}_{k^*}(\boldsymbol{\mu}_{k^*}, \boldsymbol{\Sigma}_{k^*})$ *as in Lemma 2 with two routers given by* $r : \mathbb{R}^d \rightarrow [E]$ *and* $r^{\mathrm{AC}} : \mathbb{R}^d \rightarrow [E]$ *for standard (Eqn. 2) and AC routers (Definition 2), respectively. Then the probabilities of incorrect assignments of routers* $r$ *and* $r^{\mathrm{AC}}$ *satisfy* $\Pr\left(r^{\mathrm{AC}}(\boldsymbol{h}') \neq k^*\right) \leq \Pr\left(r(\boldsymbol{h}') \neq k^*\right)$.

**Promoting faster convergence.** For an expert embedding $\boldsymbol{e}_k \in \mathbb{R}^d$ and associated cluster $\mathcal{C}_k$, our AC router in Definition 2 adaptively spheres $\mathcal{C}_k$ by stretching the feature space with weights inversely proportional to the coordinate-wise dispersion in $\mathcal{C}_k$. This reduces the conditioning number of the Hessian of the loss with respect to the expert $\boldsymbol{e}_k$, improving the loss landscape and enabling faster and more stable convergence of the router. This notion is formalized in Proposition 2:

**Proposition 2** (Faster convergence of ACMoE). *Let* $\mathcal{L}^{\mathrm{MoE}} : \boldsymbol{\Theta} \rightarrow \mathbb{R}_+$ *and* $\mathcal{L}^{\mathrm{ACMoE}} : \boldsymbol{\Theta} \rightarrow \mathbb{R}_+$ *be the network loss functions defined on the whole parameter set* $\boldsymbol{\Theta}$ *when employing the standard (Eqn. 2) and AC routers (Definition 2), respectively. Let* $\kappa(\boldsymbol{A}) = \lambda_{\max}/\lambda_{\min}$ *denote the conditioning number of a matrix* $\boldsymbol{A}$ *with largest and smallest eigenvalues* $\lambda_{\max}$ *and* $\lambda_{\min}$ *respectively. Let the Hessian of an* $i^{\mathrm{th}}$ *expert be given by* $\nabla^2_{\boldsymbol{e}_i}$. *Then for each* $i \in [E]$ *the following holds with high probability*

$$\kappa\left(\nabla^2_{\boldsymbol{e}_i}\mathcal{L}^{\mathrm{ACMoE}}\right) \leq \kappa\left(\nabla^2_{\boldsymbol{e}_i}\mathcal{L}^{\mathrm{MoE}}\right) \tag{11}$$

**Remark 4.** *Faster convergence of ACMoE can also be argued from the perspective of learning Gaussian mixture models with Expectation Maximization (Dempster et al., 1977). The classic result of Ma et al. (2000) shows the convergence rate to the true parameters depends on the overlap between component Gaussians. Our AC method adaptively transforms the input space with by* $\boldsymbol{M}_k$ *(Definition 1), which decreases component overlap by increasing inter-cluster distances.*

The proof is provided in Appendix A.3. We find this result empirically supported as shown by the rapid convergence in Fig. 2.

## 4 EXPERIMENTAL RESULTS

In this section, we empirically justify the advantage of ACMoE over baseline MoE models. We evaluate our method on large-scale tasks including Wikitext-103 (Merity et al., 2016) language modeling and ImageNet (Deng et al., 2009) object classification. We implement our AC router into Switch Transformer (Fedus et al., 2022), Generalist Language Model (GLaM) (Du et al., 2022), and Swin Transformer (Liu et al., 2021) backbones and compare our router against the standard Sparse Mixture-of-Experts (SMoE) router (Shazeer et al., 2017) and the XMoE router (Chi et al., 2022). We show that i) ACMoE obtains substantive improvements over baseline models across both language and vision tasks; ii) ACMoE offers robust improvements on contaminated and out-of-distribution samples; and iii) ACMoE attains these gains without introducing any learnable parameters and with negligible additional computational overhead. Results are averaged over 5 runs with different seeds.

### 4.1 LANGUAGE MODELING

**Experimental Setup.** We adopt the experimental setup of Pham et al. (2024). We compare AC-MoE with Switch Transformer and GLaM baselines with 16 total experts in small (70M parameters) and medium (220M parameters) configurations with top-2 expert routing. We present pretraining test perplexity (PPL) results for Wikitext-103 and test bytes-per-character (BPC) for character-level EnWik-8 . We report top-1 accuracy for finetuning classification tasks on the 2-class Stanford Sentiment Treebank-2 (SST2) (Socher et al., 2013), 5-class Stanford Sentiment Treebank-5 (SST5) (Socher et al., 2013), and 77-class Banking-77 (B77) (Casanueva et al., 2020). Full experimental details are provided in Appendix C.

Table 1: Test Perplexity (PPL) and bytes-per-character (BPC) pretraining and top-1 test accuracy on Stanford Sentiment Treebank 2, 5 (SST2, SST5), and Banking-77 (B77) finetuning classification.

| Model | Test BPC / PPL (↓) | SST2 (↑) | SST5 (↑) | B77 (↑) |
|---|---|---|---|---|
| *EnWik-8 Pretrain* | | | | |
| *Switch Transformer* (Fedus et al., 2022) | 1.153 | 63.27 | 32.21 | 53.48 |
| Switch-ACMoE (**Ours**) | **1.137** | **64.45** | **33.79** | **54.26** |
| *WikiText-103 Pretrain* | | | | |
| *Switch Transformer* (Fedus et al., 2022) | 35.48 | 76.27 | 39.13 | 83.82 |
| Switch-ACMoE (**Ours**) | **34.42** | **77.32** | **40.04** | **86.01** |
| *GLaM* (Du et al., 2022) | 38.27 | 69.97 | 33.69 | 80.89 |
| GLaM-ACMoE (**Ours**) | **36.26** | **71.90** | **34.24** | **82.33** |

Table 2: Perplexity (PPL) on WikiText-103 contaminated by Text Attack.

| Model | Clean Test PPL (↓) | Contaminated Test PPL (↓) |
|---|---|---|
| *Switch Transformer* (Fedus et al., 2022) | 35.48 | 48.12 |
| Switch-ACMoE (**Ours**) | **34.42** | **47.61** |
| *GLaM* (Du et al., 2022) | 38.27 | 50.84 |
| GLaM-ACMoE (**Ours**) | **36.26** | **47.91** |

**Pretraining and Finetuning.** Table 3 shows ACMoE attains top test PPL on WikiText-103 language modeling in Switch and GLaM backbones at small and medium configurations under baseline SMoE and XMoE routers. The improvement in the GLaM-medium architecture is a particularly substantive 4.8% over the next best baseline. Table 1 shows ACMoE pretrained models on both WikiText-103 and EnWik-8 surpass the performance of baselines in finetuning tasks, with strong, consistent improvements of approximately 3%, showing ACMoE's strong performance carries over to finetuning.

**Robust Language Modeling.** Table 2 show test PPL on WikiText-103 contaminated by Text Attack, where words are randomly swapped with a generic token 'AAA'. We follow the setup of Han et al. (2024); Teo & Nguyen (2025b); Abdullaev & Nguyen (2025) and assess models by training them on clean data before attacking the test data using an attack rate of 2.5%. ACMoE outperforms baseline Switch and GlaM with particularly robust performance in the GLaM backbone, surpassing GLaM by 5.8%.

### 4.2 IMAGE CLASSIFICATION

**Experimental Setup.** We adopt the experimental setup of Liu et al. (2021) for pretraining and evaluation on ImageNet. We evaluate ACMoE against the Swin Transformer baseline with 16 total experts in both top-1 and top-2 expert routing settings. The Swin backbone has 280M parameters. We additionally conduct experiments on ImageNet under white box adversarial attacks

Table 3: WikiText-103 test PPL of ACMoE and baseline GLaM and Switch.

| Router | Test PPL (↓) |
|---|---|
| *Switch Transformer* (Fedus et al., 2022) | |
| *SMoE-small* (Shazeer et al., 2017) | 87.94 |
| *XMoE-small* (Chi et al., 2022) | 87.21 |
| ACMoE-small (**Ours**) | **85.07** |
| *SMoE-medium* (Shazeer et al., 2017) | 35.48 |
| *XMoE-medium* (Chi et al., 2022) | 35.88 |
| *StableMoE-medium* (Dai et al., 2022) | 35.33 |
| ACMoE-medium (**Ours**) | **34.42** |
| *GLaM* (Du et al., 2022) | |
| *SMoE-small* (Shazeer et al., 2017) | 58.27 |
| *XMoE-small* (Chi et al., 2022) | 54.80 |
| ACMoE-small (**Ours**) | **54.55** |
| *SMoE-medium* (Shazeer et al., 2017) | 38.27 |
| *XMoE-medium* (Chi et al., 2022) | 38.10 |
| *StableMoE-medium* (Dai et al., 2022) | 38.04 |
| ACMoE-medium (**Ours**) | **36.26** |

fast gradient sign method (FGSM) (Goodfellow et al., 2014) and projected gradient descent (PGD) (Madry et al., 2017), and black box attack simultaneous perturbation stochastic approximation (SPSA) (Uesato et al., 2018). We also present results on out-of-distribution (OOD)(Hendrycks et al., 2021a;b). In all robust image classification tasks, image classification using ImageNet-A/O/R we adopt the conventional setup of pretraining on ImageNet and evaluating the trained models on the contaminated/OOD datasets (Han et al., 2024; Zhou et al., 2022a; Puigcerver et al., 2022; Nguyen et al., 2024; Nielsen et al., 2025). Full experimental details are provided in Appendix C.

Table 4: Test Accuracy on ImageNet corrupted PGD, FGSM, and SPSA.

| Model | Clean Data | | PGD | | FGSM | | SPSA | |
|---|---|---|---|---|---|---|---|---|
| | Top 1 | Top 5 | Top 1 | Top 5 | Top 1 | Top 5 | Top 1 | Top 5 |
| *Swin-Top 1* (Liu et al., 2021) | 75.22 | 92.51 | 39.69 | **74.59** | 52.84 | **83.86** | 59.92 | **82.63** |
| Swin-ACMoE-Top 1 (**Ours**) | **75.39** | **92.56** | **40.66** | 73.46 | **53.43** | 82.80 | **59.97** | 82.47 |
| *Swin-Top 2* (Liu et al., 2021) | 76.10 | 92.99 | 40.85 | 75.51 | 54.70 | 85.22 | 60.57 | 82.75 |
| Swin-ACMoE-Top 2 (**Ours**) | **76.31** | **93.14** | **43.74** | **78.55** | **55.78** | **85.80** | **63.47** | **86.05** |

Table 5: Test Accuracy on Image Classification in Imagenet-A/O/R

| Model | Im-A Top-1 Acc. ($\uparrow$) | Im-R Top-1 Acc. ($\uparrow$) | Im-O AUPR ($\uparrow$) |
|---|---|---|---|
| *Swin Transformer-Top 1* (Liu et al., 2021) | 6.83 | 30.60 | 17.89 |
| Swin-ACMoE-Top 1 (**Ours**) | **7.13** | **30.85** | **18.45** |
| *Swin Transformer-Top 2* (Liu et al., 2021) | 9.38 | 32.07 | 18.51 |
| Swin-ACMoE-Top 2 (**Ours**) | **9.42** | **32.35** | **19.55** |

**Image Classification under Adversarial Attack.** Table 4 shows performance on ImageNet classification against FGSM, PGD, and SPSA. Compared with Swin baseline, ACMoE-Top 2 attains noteworthy 7% and 5% improvements against PGD and SPSA in top-1 accuracy respectively.

**Out-of-distribution Image Classification.** Table 5 shows ACMoE improves over the baseline Swin Transformer in image classification on hard OOD and real-world adversarially filtered images. Evaluation on ImageNet-A/O/R shows consistent improvements over the baseline in top-1 and top-2 expert choice, with particularly strong improvements in ImageNet-O under top-2 routing with a performance gain in area under precision recall (AUPR) of almost 6%.

### 4.3 EMPIRICAL ANALYSIS

**Load Balancing.** We analyze in Table 6 the effect of ACMoE on expert load balancing. Load balance is calculated as the percentage of tokens assigned to each expert. The load balance score is then taken as the standard deviation over these percentages. A standard deviation of 0, where all experts are activated in exactly equal proportions, is therefore a perfect load balance. We compute this statistic per MoE layer and present the overall load balance averaged over all layers. ACMoE attains better overall load balancing compared to Switch and Swin transformers. Against all backbones, ACMoE achieves a smaller spread in the load balances over layers, shown by smaller standard deviation. Visually we see how better expert specialization can aid load balance in Fig. 1, where better identification of the semantic regions of the input space leads to more experts being activated.

**Efficiency Analysis.** Computing the cluster-wise feature weights $\{w_k\}_{k \in [E]}$ requires no learnable parameters and is obtained by computing the mean absolute deviation for each set of tokens assigned to the $k^{\text{th}}$ expert. This can be computed using just two computations of the mean – one for the mean per cluster and one for the mean of the absolute deviations per cluster – done in parallel over all clusters. This is of order $\mathcal{O}(2nd) = \mathcal{O}(n)$ for $n$ tokens, hence the upper-bound time complexity of the MoE layer is unaffected. Table 7 provides empirical efficiency analysis in terms of compute speed, memory allocation, and parameters, which shows changes in speed and memory are within a margin of approximately 1% or less, implying there is no significant efficiency loss.

## 5 RELATED WORK

**Routing Methods.** Recent studies have proposed token-expert assignment algorithms based on reinforcement learning (Bengio et al., 2015), deterministic hashing (Roller et al., 2021), optimal transport (Liu et al., 2022), linear programs (Lewis et al., 2021), cosine similarity (Chi et al., 2022), soft token mixing (Puigcerver et al., 2023), greedy top-k experts per token (Shazeer et al., 2017) and greedy top-k tokens per expert (Zhou et al., 2022b). Existing work has predominantly considered dot-products between inputs and experts as a suitable metric for similarity (Lewis et al., 2021; Puigcerver et al., 2023; Shazeer et al., 2017; Zhou et al., 2022b; Chi et al., 2022). This work continues with dot-product based learnable routing but computes the routing assignments in an adaptively transformed space to maximally identify the latent expert clusters.

Table 6: Load Balance Analysis of ACMoE and Baseline MoE Models

| Model | Layer-Averaged Load Balance ($\downarrow$) |
|---|---|
| *Switch Transformer* (Fedus et al., 2022) | $5.577 \pm 4.131$ |
| Switch-ACMoE (**Ours**) | $\mathbf{5.317} \pm \mathbf{2.622}$ |
| *GLaM* (Du et al., 2022) | $\mathbf{2.901} \pm 1.434$ |
| GLaM-ACMoE (**Ours**) | $2.938 \pm \mathbf{1.221}$ |
| *Swin Transformer* (Liu et al., 2021) | $2.134 \pm 1.110$ |
| Swin-ACMoE (**Ours**) | $\mathbf{2.127} \pm \mathbf{0.968}$ |

Table 7: Efficiency Comparison between ACMoE and baseline MoE Models

| Model | Compute Speed (ms/it) | Max Memory (K) | #Params (M) |
|---|---|---|---|
| *GLaM* (Du et al., 2022) | 422.62 | 25.69 | 220 |
| GLaM-ACMoE (**Ours**) | 425.15 | 25.72 | 220 |
| *Switch Transformer* (Fedus et al., 2022) | 391.93 | 34.64 | 216 |
| Switch-ACMoE (**Ours**) | 393.29 | 34.68 | 216 |
| *Swin Transformer* (Liu et al., 2021) | 403.36 | 22.00 | 280 |
| Swin-ACMoE (**Ours**) | 408.56 | 22.19 | 280 |

**MoE and Cluster Analysis.** The MoE framework traces its roots back to Gaussian mixture models where the input space is assumed divisible into separate regions with an expert specializing in each region (Jacobs et al., 1991). Recent studies show that the router can recover the clustering structure of the input space and each expert specializes in a specific cluster (Dikkala et al., 2023; Chen et al., 2022). Our work leverages the clustering perspective on MoE to consider adaptive transformations of the input space to more easily distinguish latent clusters. We learn these transformations via feature-weighted cluster analysis, which has been studied in the clustering literature (Brusco & Cradit, 2001; Witten & Tibshirani, 2010; Gnanadesikan et al., 1995; Van Buuren & Heiser, 1989; Friedman & Meulman, 2004). Friedman & Meulman (2004) consider cluster-dependent feature weights to augment iterative clustering algorithms. Our approach similarly uses cluster-dependent feature weights but uses a different optimization problem to derive optimal weights.

**Robust MoE.** The robustness of MoE architectures is a newly emerging research area. Puigcerver et al. (2022) provide the first study in this direction from the perspective of model capacity and the Lipschitz constant, finding conditions under which MoE models are provably more robust than their dense counterparts. Zhang et al. (2023) examine the effect of adversarial training and propose an alternating optimization adversarial defence. Teo & Nguyen (2024) integrates heavy-ball momentum in SMoE to improve the model's stability and robustness. Our work differs from these approaches by examining the robustness of MoE models purely through the lens of the latent clustering structure of the input space. To the best of our knowledge, this is a novel lens on robustness in MoE models.

## 6 CONCLUSION AND FUTURE WORK

In this paper, we present the Adaptive Clustering (AC) router and ACMoE layer, a novel MoE routing method that computes token-expert assignments in a transformed space that maximally identifies latent clusters in the data and more easily discovers the best-matched expert for each token. We adaptively learn for each input which features are relevant to determining its latent cluster assignment and scale its features accordingly such that features that promote tight clustering are upweighted and features that produce dispersed clusters are downweighted. This transformation accentuates the relevant characteristics of each input according to the specialization of the experts, thereby allowing the router to more easily discover the optimal input-expert allocation. Our AC routing method enables faster convergence by improving the Hessian conditioning of the router and better robustness by increasing the separation of latent clusters in the transformed space. This approach makes no assumptions on the downstream task, requires no learnable parameters, and can be applied within any MoE architecture to boost performance on clean and contaminated data. A limitation of our method is that the AC router requires estimates of each token's cluster assignment. We obtain these by using the expert assignments in previous layers, which means we require the embedding size to remain the same between adjacent MoE layers. For ongoing work, we are investigating improved methods for estimating the latent cluster memberships without reliance on previous layers and with provable consistency guarantees.

## ACKNOWLEDGMENTS AND DISCLOSURE OF FUNDING

This research / project is supported by the National Research Foundation Singapore under the AI Singapore Programme (AISG Award No: AISG2-TC-2023-012-SGIL). This research / project is supported by the Ministry of Education, Singapore, under the Academic Research Fund Tier 1 (FY2023) (A-8002040-00-00, A-8002039-00-00). This research / project is also supported by the NUS Presidential Young Professorship Award (A-0009807-01-00).

Thanks to our anonymous reviewers, who provided valuable feedback which improved the paper substantially. Thanks also to Loi Xuan Ly for lending his eye for design.

**Reproducibility Statement.** Source code for our experiments are provided in the supplementary material. We provide the full details of our experimental setup – including datasets, model specification, train regime, and evaluation protocol – for all experiments in Appendix C. All datasets are publicly available.

**Ethics Statement.** Our work considers fundamental architectures, and in particular their robustness and convergence properties. Given this, we foresee no issues regarding fairness, privacy, or security, or any other harmful societal or ethical implications in general.

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

# Supplement to "Tight Clusters Make Specialized Experts"

**Table of Contents**

## A  TECHNICAL PROOFS

### A.1  PROOF OF THEOREM 1

To begin with, we present the following lemma to show the existence of constants $\alpha_k$ for $k \in [E]$ that satisfy Eqn. 7:

**Lemma 3.** *For any $\lambda > 0$, Eqn. 7 has exactly $d$ real solutions with respect to $\alpha_k$.*

*Proof of Lemma 3.* Without loss of generality, assume that $s_{1k} \geq s_{2k} \geq \cdots \geq s_{dk}$. Denote

$$\varphi(\alpha) := \sum_{q \in [d]} \frac{1}{s_{qk} + \alpha} - \frac{d}{\lambda}. \tag{12}$$

Then, the existence of solutions to Eqn. 7 is equivalent to the condition $\varphi(\alpha_l) = 0$. Note that $\varphi(\alpha)$ is a strictly decreasing function in its connected continuity domains since

$$\varphi'(\alpha) = - \sum_{q \in [d]} \frac{1}{(s_{qk} + \alpha)^2} < 0 \tag{13}$$

for all $\alpha \in \mathbb{R} \setminus \{-s_{1k}, \ldots, -s_{dk}\}$. Further, we observe that

$$\lim_{\alpha \to -s_{qk}^-} \varphi(\alpha) = -\infty, \qquad \lim_{\alpha \to -s_{qk}^+} \varphi(\alpha) = +\infty \tag{14}$$

for all $q \in [d]$, and

$$\lim_{\alpha \to \pm\infty} \varphi(\alpha) = -\frac{d}{\lambda} < 0. \tag{15}$$

Now consider the domain of continuity of $\varphi(\alpha)$, namely $(-\infty, -s_{1k}) \cup (-s_{1k}, -s_{2k}) \cup \cdots \cup (-s_{dk}, \infty)$. Due to the monotonicity and limits 14 & 15, there exists a unique solution in each of the intervals except for $(-\infty, -s_{1k})$ where the function is always strictly negative, thus, yielding $d$ roots in total. $\quad\square$

Now we follow up with the main proof of this section.

*Proof of Theorem 1.* First, let $\mathcal{I}_k := \{i : r(i) = k\}$ for convenience. Now let us restate the clustering optimization problem (4) here once again:

$$\min_{\boldsymbol{w}_k} Q(c, \{\boldsymbol{w}_k\}_{k \in [E]}) = \sum_{k \in [E]} \frac{1}{N_k^2} \sum_{i,j \in \mathcal{I}_k} \sum_{q \in [d]} \left( w_{qk} \rho_{ijq} + \frac{\lambda}{d} \log \frac{1}{d w_{qk}} \right),$$

$$\text{such that } \sum_{q \in [d]} w_{qk} = 1, \quad \forall k \in [E], \tag{16}$$

where we have immediately used the fact that

$$D_{\mathrm{KL}}(\boldsymbol{u} \parallel \boldsymbol{w}_k) = \sum_{q \in [d]} \frac{1}{d} \log \frac{1/d}{w_{qk}}. \tag{17}$$

Also, note that

$$\sum_{q \in [d]} \left( w_{qk} \rho_{ijq} + \lambda \frac{1}{d} \log \frac{1}{d w_{qk}} \right) = \sum_{q \in [d]} \left( w_{qk} \rho_{ijq} - \lambda \frac{1}{d} \log(d w_{qk}) \right)$$

$$= \sum_{q \in [d]} \left( w_{qk} \rho_{ijq} - \frac{\lambda}{d} \log w_{qk} \right) - \lambda \log d. \tag{18}$$

We can ignore the term $\lambda \log d$ since it does not depend on the optimization variable. Method of Lagrange multipliers turns this constrained optimization problem into the following unconstrained counterpart:

$$\min_{\boldsymbol{w}_k, \boldsymbol{\alpha}} \mathcal{L}(c, \{\boldsymbol{w}_k\}_{k \in [E]}, \boldsymbol{\alpha}) = \sum_{k \in [E]} \frac{1}{N_k^2} \sum_{i,j \in \mathcal{I}_k} \sum_{q \in [d]} \left( w_{qk} \rho_{ijq} - \frac{\lambda}{d} \log w_{qk} \right) + \sum_{k \in [E]} \alpha_k \left( \sum_{q \in [d]} w_{qk} - 1 \right),$$

where $\boldsymbol{\alpha} = \begin{bmatrix} \alpha_1 & \cdots & \alpha_L \end{bmatrix}^\top$ is the vector of Lagrange multipliers. Note that the last optimization problem can be separated into the following $L$ independent optimization subproblems:

$$\min_{\boldsymbol{w}_k, \boldsymbol{\alpha}} \mathcal{L}_k(c, \boldsymbol{w}_k, \boldsymbol{\alpha}) = \frac{1}{N_k^2} \sum_{i,j \in \mathcal{I}_k} \sum_{q \in [d]} \left( w_{qk} \rho_{ijq} - \frac{\lambda}{d} \log w_{qk} \right) + \alpha_k \left( \sum_{q \in [d]} w_{qk} - 1 \right),$$

for $k \in [E]$. Since the objective function is a positive combination of convex functions, the optimization problem is also convex. By setting the derivatives of $\mathcal{L}_k$ with respect to both optimization variables to 0, we obtain the following system of equations:

$$
\begin{cases}
\dfrac{\partial \mathcal{L}_k}{\partial w_{qk}} = s_{qk} - \dfrac{\lambda}{d} \dfrac{1}{w_{qk}} + \alpha_k = 0, \\
\dfrac{\partial \mathcal{L}_k}{\partial \alpha_k} = \sum_{q \in [d]} w_{qk} - 1 = 0
\end{cases}
$$

for all $k \in [E]$, where $s_{qk}$ is the data dispersion measure defined in the theorem statement. The first equation yields

$$
w_{qk} = \frac{\lambda}{d} \frac{1}{s_{qk} + \alpha_k}, \tag{19}
$$

where $\alpha_k$ is found from $\sum_{q \in [d]} w_{qk} = 1$ which in fact gives

$$
\sum_{q \in [d]} \frac{1}{s_{qk} + \alpha_k} = \frac{d}{\lambda} \tag{20}
$$

for all $k \in [E]$ as desired. $\qquad\square$

## A.2 PROOF OF PROPOSITION 1

Since Proposition 1 is a composition of Lemma 1 and Lemma 2, we proceed by providing their proofs.

### A.2.1 PROOF OF LEMMA 1

*Proof of Lemma 1.* Notice that we can expand inequality (1) as

$$
\sum_{i \in [d]} m_i \delta \mu_i^2 \geq \sum_{i \in [d]} \delta \mu_i^2,
$$

where we let $\delta \boldsymbol{\mu} := \boldsymbol{\mu}_b - \boldsymbol{\mu}_a$. Since $\boldsymbol{M}_a$ entries are mean-scaled, we can rewrite them as

$$
m_i = \frac{d m_i'}{\sum_{j \in [d]} m_j'} \tag{21}
$$

for some initial dispersion estimates $\{m_j'\}_{j \in [d]}$. Without loss of generality, assume that $[d']$ is the set of dimension indices for which the dispersions are relatively much smaller than those in the rest of the dimensions in the sense that $m_i' \gg m_j'$ for any $i \in [d']$ and $j \in [d] \smallsetminus [d']$. Then, there exists a positive $\alpha \ll 1/2$ such that $\sum_{i \in [d']} m_i > d - \alpha$ and $\sum_{i \in [d] \smallsetminus [d']} m_i < \alpha$. By the assumption that clusters are best-separated along the features for which they cluster tightly, this means that the weight matrix $\boldsymbol{M}_a$ maximizes the contribution of largest $d'$ terms in $\sum_{i \in [d]} m_i \delta \mu_i^2$ corresponding to individual feature-wise distances in dimensions where the feature dispersions are the smallest instead of giving uniform weights to all dimensions, which leads to inequality (1). $\qquad\square$

### A.2.2 PROOF OF LEMMA 2

*Proof of Lemma 2.* Since we use the $\mathcal{L}_2$ distance between the token $\boldsymbol{h}$ and $\boldsymbol{\mu}_c$ as a similarity metric, we assign cluster $g_{k^*}$ to the token $\boldsymbol{h}'$ iff $\|\boldsymbol{h}' - \boldsymbol{\mu}_{k^*}\| \leq \|\boldsymbol{h}' - \boldsymbol{\mu}_k\|$. Assume that the token $\boldsymbol{h}'$ is a noisy observation of an underlying true token $\boldsymbol{h}$ which actually originates from cluster $g_{k^*}$. Then, the token $\boldsymbol{h}'$ can be decomposed as $\boldsymbol{h}' = \boldsymbol{h} + \boldsymbol{\epsilon}$ for a random noise $\boldsymbol{\epsilon} \sim \mathcal{N}(\boldsymbol{0}, \boldsymbol{\Sigma}_\epsilon)$. Now define the decision variable $\mathcal{D}(\boldsymbol{h}') := \|\boldsymbol{h}' - \boldsymbol{\mu}_{k^*}\|^2 - \|\boldsymbol{h}' - \boldsymbol{\mu}_k\|^2$ which turns the clustering condition to $\mathcal{D}(\boldsymbol{h}') \leq 0$ for the cluster $g_{k^*}$. Let us analyze the decision variable $\mathcal{D}$ as a random variable where randomness may come from the underlying sampling strategy and noise. Note that

$$
\begin{aligned}
\mathcal{D}(\boldsymbol{h}') &= \|\boldsymbol{h} + \boldsymbol{\epsilon} - \boldsymbol{\mu}_{k^*}\|^2 - \|\boldsymbol{h} + \boldsymbol{\epsilon} - \boldsymbol{\mu}_k\|^2 \\
&= \|\boldsymbol{h} - \boldsymbol{\mu}_{k^*}\|^2 - \|\boldsymbol{h} - \boldsymbol{\mu}_k\|^2 + 2(\boldsymbol{\mu}_k - \boldsymbol{\mu}_{k^*})^\top \boldsymbol{\epsilon} \\
&= \mathcal{D}(\boldsymbol{h}) + 2 \delta \boldsymbol{\mu}^\top \boldsymbol{\epsilon}, \tag{22}
\end{aligned}
$$

where $\delta\boldsymbol{\mu} := \boldsymbol{\mu}_k - \boldsymbol{\mu}_{k^*}$. Due to the assumption that $\boldsymbol{h}$ is drawn from the distribution $g_{k^*}$, it can be rewritten as $\boldsymbol{h} = \boldsymbol{\mu}_{k^*} + \boldsymbol{\nu}$ with $\boldsymbol{\nu} \sim \mathcal{N}(\mathbf{0}, \boldsymbol{\Sigma}_{k^*})$. Then for the first term in Eqn. 22, we have

$$
\begin{aligned}
\mathcal{D}(\boldsymbol{h}) &= \|\boldsymbol{h} - \boldsymbol{\mu}_{k^*}\|^2 - \|\boldsymbol{h} - \boldsymbol{\mu}_k\|^2 \\
&= \delta\boldsymbol{\mu}^\top (2\boldsymbol{h} - \boldsymbol{\mu}_{k^*} - \boldsymbol{\mu}_k) \\
&= \delta\boldsymbol{\mu}^\top (2\boldsymbol{\nu} - \delta\boldsymbol{\mu}) \\
&= 2\delta\boldsymbol{\mu}^\top \boldsymbol{\nu} - \|\delta\boldsymbol{\mu}\|^2 .
\end{aligned}
\tag{23}
$$

Substituting this back into Eqn. 22, we get

$$
\mathcal{D}(\boldsymbol{h}') = 2\delta\boldsymbol{\mu}^\top (\boldsymbol{\nu} + \boldsymbol{\epsilon}) - \|\delta\boldsymbol{\mu}\|^2 .
\tag{24}
$$

This shows that $\mathcal{D}(\boldsymbol{h}') \sim \mathcal{N}\left(-\|\delta\boldsymbol{\mu}\|^2, 4\delta\boldsymbol{\mu}^\top (\boldsymbol{\Sigma}_{k^*} + \boldsymbol{\Sigma}_\epsilon)\delta\boldsymbol{\mu}\right)$. Since $\mathcal{D}(\boldsymbol{h}')$ follows a normal distribution with the derived parameters, the probability that $\boldsymbol{h}'$ is assigned to cluster $g_{k^*}$ is given by

$$
\Pr(\text{correct cluster}) = \Pr\left(\mathcal{D}(\boldsymbol{h}) \le 0\right) = \Phi\left(\frac{\|\delta\boldsymbol{\mu}\|^2}{2\sqrt{\delta\boldsymbol{\mu}^\top (\boldsymbol{\Sigma}_{k^*} + \boldsymbol{\Sigma}_\epsilon)\delta\boldsymbol{\mu}}}\right),
\tag{25}
$$

where $\Phi$ denotes the CDF of normal distribution as usual. Since $\Phi$ is an increasing function, the probability that the noisy token $\boldsymbol{h}$ is assigned to the correct cluster is proportional to the distance between the cluster centroids and inverse proportional to the covariance matrices of the cluster and the additive noise. On the other hand, for the incorrect clustering probability, we have

$$
\Pr(\text{incorrect cluster}) = 1 - \Phi\left(\frac{\|\delta\boldsymbol{\mu}\|^2}{2\sqrt{\delta\boldsymbol{\mu}^\top (\boldsymbol{\Sigma}_{k^*} + \boldsymbol{\Sigma}_\epsilon)\delta\boldsymbol{\mu}}}\right)
\tag{26}
$$

as claimed. $\qquad\square$

### A.3 PROOF OF PROPOSITION 2

*Proof of Proposition 2.* Let the router be given by $g$ and let the softmax function be given by $g_{\boldsymbol{\theta}} : \mathbb{R}^d \to \mathbb{R}^d$, parameterized by expert embeddings $\{\boldsymbol{e}_i\}_{i \in [E]}$. The network loss depends on expert embeddings only through the router function $g$. We shall explore the exclusive contribution of each expert embedding in minimizing $\mathcal{L}^{\text{ACMoE}}$. In order to do this, we look at the network loss as a scalar function of $i^{\text{th}}$ expert embedding vector while treating all other network parameters as fixed. Then, we can write $\mathcal{L}^{\text{ACMoE}} : \mathbb{R}^d \to \mathbb{R}$ such that $\mathcal{L}^{\text{ACMoE}} = \mathcal{L}^{\text{ACMoE}}(g_{\boldsymbol{\theta}}(\boldsymbol{e}_i))$. For simplicity, we shall omit the subscript $\boldsymbol{\theta}$. The gradient that comes from back-propagation is then given by

$$
\nabla_{\boldsymbol{e}_i} \mathcal{L}^{\text{ACMoE}} = \left(\nabla_g \mathcal{L}^{\text{ACMoE}}\right)^\top \nabla_{\boldsymbol{e}_i} g,
\tag{27}
$$

where $\nabla_{\boldsymbol{e}_i} g \in \mathbb{R}^{d \times d}$ denotes the Jacobian matrix of $g$ since for $g_k := (g_{\boldsymbol{\theta}}(\boldsymbol{e}_i))_k$, we can write

$$
\frac{\partial}{\partial e_{is}} \mathcal{L}^{\text{ACMoE}}(g_1, \ldots, g_d) = \sum_k \frac{\partial \mathcal{L}^{\text{ACMoE}}}{\partial g_k} \frac{\partial g_k}{\partial e_{is}} .
\tag{28}
$$

Note that for $g_k = \text{softmax}(\boldsymbol{h}^\top \boldsymbol{M} \boldsymbol{e}_k)$, we have

$$
\frac{\partial g_k}{\partial e_{is}} = m_s h_s g_k (\delta_{ki} - g_i) = m_s h_s b_{ki} .
\tag{29}
$$

Then, the element of the Hessian matrix of the network loss at index $(s, t) \in [d] \times [d]$ can be written as

$$
\begin{aligned}
\boldsymbol{H}_{st}^{(i)}(\mathcal{L}^{\text{ACMoE}}) &= \frac{\partial^2 \mathcal{L}^{\text{ACMoE}}}{\partial e_{is} \partial e_{it}} = \frac{\partial}{\partial e_{it}} \sum_k \frac{\partial \mathcal{L}^{\text{ACMoE}}}{\partial g_k} \frac{\partial g_k}{\partial e_{is}} \\
&= \sum_k \left(\sum_j \frac{\partial^2 \mathcal{L}^{\text{ACMoE}}}{\partial g_k \partial g_j} \frac{\partial g_j}{\partial e_{it}}\right) \frac{\partial g_k}{\partial e_{is}} + \frac{\partial \mathcal{L}^{\text{ACMoE}}}{\partial g_k} \frac{\partial^2 g_k}{\partial e_{is} \partial e_{it}} \\
&= m_s h_s m_t h_t \left[\sum_k \left(\sum_j \frac{\partial^2 \mathcal{L}^{\text{ACMoE}}}{\partial g_k \partial g_j} b_{ji}\right) b_{ki} + \frac{\partial \mathcal{L}^{\text{ACMoE}}}{\partial g_k} b'_{ki}\right] \\
&= m_s h_s m_t h_t B_i,
\end{aligned}
\tag{30}
$$

where $B_i$ is some constant that depends only on index $i$. Due to Eqn. 30, the Hessian takes the following matrix form

$$\boldsymbol{H}^{(i)} = B_i(\boldsymbol{M}\boldsymbol{h})(\boldsymbol{M}\boldsymbol{h})^\top. \tag{31}$$

Taking expectation from both sides, we obtain

$$\mathbb{E}_{\boldsymbol{h}\sim(\boldsymbol{\mu},\boldsymbol{\Sigma})}\left[\boldsymbol{H}^{(i)}\right] = B_i\mathbb{E}_{\boldsymbol{h}\sim(\boldsymbol{\mu},\boldsymbol{\Sigma})}\left[\boldsymbol{M}(\boldsymbol{h}\boldsymbol{h}^\top)\boldsymbol{M}\right] = B_i\boldsymbol{M}(\boldsymbol{\Sigma})\boldsymbol{M}, \tag{32}$$

where we assume $\boldsymbol{h}$ is centered. Now recall that $\boldsymbol{M} = \mathrm{diag}(m_1,\ldots,m_d)$ where for each $i$, $m_i \sim 1/\sqrt{\Sigma_{ii}}$ holds. Assume that the covariance matrix $\boldsymbol{\Sigma}$ is symmetric positive definite. Then, it is diagonalizable as $\boldsymbol{\Sigma} = \boldsymbol{U}\boldsymbol{\Lambda}\boldsymbol{U}^\top$ with $\boldsymbol{\Lambda} = \mathrm{diag}(\lambda_1,\ldots,\lambda_d)$, a diagonal matrix with eigenvalues of $\boldsymbol{\Sigma}$. With the transformation $\boldsymbol{M}$, we get

$$\boldsymbol{M}\boldsymbol{\Sigma}\boldsymbol{M} = \boldsymbol{M}\boldsymbol{U}\boldsymbol{\Lambda}\boldsymbol{U}^\top\boldsymbol{M} = \boldsymbol{U}\boldsymbol{M}\boldsymbol{\Lambda}\boldsymbol{M}\boldsymbol{U}^\top \tag{33}$$

$$= \boldsymbol{U}\begin{bmatrix} m_1^2\lambda_1 & & \\ & \ddots & \\ & & m_d^2\lambda_d \end{bmatrix}\boldsymbol{U}^\top. \tag{34}$$

Since the eigenvalues capture the variances along the principal components of the covariance matrix, $m_i^2$, as a reciprocal of a measure of dimension-wise dispersion, is reasonably correlated with $1/\lambda_i$, as demonstrated by Lemma 4, implying $\lambda_j \le \lambda_i \implies m_j \ge m_i$ with high probability. Therefore, we obtain that

$$\kappa(\boldsymbol{M}\boldsymbol{\Sigma}\boldsymbol{M}) = \frac{\lambda_{\max}(\boldsymbol{M}\boldsymbol{\Sigma}\boldsymbol{M})}{\lambda_{\min}(\boldsymbol{M}\boldsymbol{\Sigma}\boldsymbol{M})} \approx \frac{m_{\min}^2\lambda_{\max}(\boldsymbol{\Sigma})}{m_{\max}^2\lambda_{\min}(\boldsymbol{\Sigma})} \le \kappa(\boldsymbol{\Sigma}), \tag{35}$$

which implies the claim. $\qquad\square$

**Lemma 4** (Correlation between dimension-wise varainces and covariance eigenvalues). *Let $\{\boldsymbol{b}_i\}_{i\in d}$ be the set of normalized basis vectors of $\mathbb{R}^d$. Consider a symmetric positive definite covariance matrix $\boldsymbol{\Sigma}$ and its unit eigenvectors $\{\boldsymbol{v}_i\}_{i\in[d]}$. Assume that the eigenvector $\boldsymbol{v}_i$ is a reasonably small perturbation of the basis vector $\boldsymbol{b}_i$ such that $\boldsymbol{v}_i^\top\boldsymbol{b}_i \ge 1 - \epsilon$ for all $i \in [d]$ and a small constant $\epsilon > 0$. Then, for all $i \in [d]$, we have*

$$|\lambda_i - \Sigma_{ii}| \le \epsilon \cdot \max_{j\ne i}|\lambda_i - \lambda_j|, \tag{36}$$

*where $\{\lambda_i\}_{i\in[d]}$ is the set of ordered eigenvalues of $\boldsymbol{\Sigma}$ corresponding to eigenvectors $\{\boldsymbol{v}_i\}_{i\in[d]}$.*

*Proof of Lemma 4.* Note that each diagonal element of the SPD covariance matrix $\boldsymbol{\Sigma}$ can be written as

$$\Sigma_{ii} = \boldsymbol{b}_i^\top\boldsymbol{\Sigma}\boldsymbol{b}_i = \boldsymbol{b}_i^\top\left(\sum_{j\in[d]}\lambda_j\boldsymbol{v}_j\boldsymbol{v}_j^\top\right)\boldsymbol{b}_i = \sum_{j\in[d]}\lambda_j(\boldsymbol{v}_j^\top\boldsymbol{b}_i)^2. \tag{37}$$

Then, the difference on the left hand side of Eqn. 36 can be bounded as

$$|\lambda_i - \Sigma_{ii}| = \left|\lambda_i - \sum_{j\in[d]}\lambda_j(\boldsymbol{v}_j^\top\boldsymbol{b}_i)^2\right| = \left|\lambda_i\left(1 - (\boldsymbol{v}_i\boldsymbol{e}_i)^2\right) - \sum_{j\ne i}\lambda_j(\boldsymbol{v}_j^\top\boldsymbol{b}_i)^2\right|$$

$$= \left|\lambda_i\sum_{j\ne i}(\boldsymbol{v}_j^\top\boldsymbol{b}_i)^2 - \sum_{j\ne i}\lambda_j(\boldsymbol{v}_j^\top\boldsymbol{b}_i)^2\right| \tag{38}$$

$$= \left|\sum_{j\ne i}(\lambda_i - \lambda_j)(\boldsymbol{v}_j^\top\boldsymbol{b}_i)^2\right|$$

$$\le \max_{j\ne i}|\lambda_i - \lambda_j|\sum_{j\ne i}(\boldsymbol{v}_j^\top\boldsymbol{b}_i)^2$$

$$= \max_{j\ne i}|\lambda_i - \lambda_j|\left(1 - (\boldsymbol{v}_i\boldsymbol{b}_i)^2\right) \tag{39}$$

$$\le \epsilon\max_{j\ne i}|\lambda_i - \lambda_j|,$$

where we used the fact that

$$\sum_{j\in[d]}(\boldsymbol{v}_j^\top\boldsymbol{b}_i)^2 = \left(\sum_{j=1}^n\left(\boldsymbol{v}_j^\top\boldsymbol{b}_i\right)\boldsymbol{v}_j\right)^\top\left(\sum_{k=1}^n\left(\boldsymbol{v}_k^\top\boldsymbol{b}_i\right)\boldsymbol{v}_k\right) = \boldsymbol{b}^\top\boldsymbol{b} = 1$$

to obtain Eqn. 38 and Eqn. 39 since the eigenvectors of $\boldsymbol{\Sigma}$ are orthonormal. $\qquad\square$

## B  IMPLEMENTATION PROCEDURE AND COMPUTATIONAL EFFICIENCY

**Training and Inference.**   Given the AC routing scheme requires requires the expert assignment per token from the previous layer, we can only implement AC routing from the second layer on. We incorporate AC routing into both training and inference stages. This is because, firstly, AC routing is designed to offer improvements to both clean and contaminated data, and so even in the presence of completely clean train and test data, it is advantageous to incorporate the AC method into both stages. Secondly, it is commonplace to encounter data contamination only at the test stage and indeed highly possible to encounter it in train as well. Therefore, in the interest of robustness as well, AC routing is incorporated into both stages.

**Computational Efficiency.**   Computing the required $\{\boldsymbol{w}_k\}_{k\in[E]}$ for number of experts $E$ requires no learnable parameters and is obtained simply by computing the mean absolute deviation for each set of tokens assigned to the $k^{\text{th}}$ expert. This can be computed using just two computations of the mean – once for the mean per cluster and once again for the mean of the absolute deviations per cluster – done in parallel over all clusters using `torch.index_reduce()` and is of the order $\mathcal{O}(2nd) = \mathcal{O}(n)$ for $n$ tokens. Hence the upper-bound time complexity of the MoE layer is unaffected. We provide in Table 7 additional efficiency analysis in terms of throughput, max GPU memory allocated, and parameters which shows no significant efficiency loss compared to baseline MoE architectures.

## C  EXPERIMENTAL DETAILS AND ADDITIONAL EXPERIMENTS

### C.1  LANGUAGE MODELING

#### C.1.1  DATASETS

**WikiText-103.**   The WikiText-103[3] dataset contains around 268K words and its training set consists of about 28K articles with 103M tokens. This corresponds to text blocks of about 3600 words. The validation set and test sets consist of 60 articles with 218K and 246K tokens respectively.

**EnWik-8.**   The EnWik-8 dataset is a byte-level dataset of 100 million bytes derived from Wikipedia that, in addition to English text, also includes markup, special characters, and text in other languages. EnWik-8 contains 90M characters for training, 5M for validation, and 5M for testing.

**Stanford Sentiment Treebank-2.**   The Stanford Sentiment Treebank-2 (SST2) (Socher et al., 2013) is a 2 class corpus with fully labeled parse trees for analysis of the compositional effects of sentiment in language. The dataset consists of 11,855 single sentences extracted from movie reviews. It was parsed with the Stanford parser and includes 215,154 unique phrases from the parse trees, each annotated by 3 human judges.

**Stanford Sentiment Treebank-5.**   Stanford Sentiment Treebank-5 (SST5) (Socher et al., 2013) is a 5 class dataset used for sentiment analysis. It consists of 11,855 single sentences extracted from movie reviews. It includes 215,154 unique phrases from parse trees, each annotated by 3 human judges. Phrases are classified as negative, somewhat negative, neutral, somewhat positive, or positive.

**Banking-77.**   Banking-77 (B77) (Casanueva et al., 2020) is a highly fine-grained 77 class classification dataset comprising 13083 customer service queries labelled with 77 intents.

#### C.1.2  MODEL, OPTIMIZER, & TRAIN SPECIFICATION

**Models.**   We use as backbones the Switch Transformer (Fedus et al., 2022) and Generalist Language Model (Du et al., 2022). Table 8 contains the specification over self-attention (SA) layers, feed-forward network (FFN) layers, Mixture-of-Experts (MoE) layers, attention span (Att. Span),

---

[3]www.salesforce.com/products/einstein/ai-research/the-wikitext-dependency-language-modeling-dataset/

embedding size and parameter count for both backbones at small and medium configurations for each pretraining task. All backbones use 16 experts with top-2 expert routing.

Table 8: Language Modeling Backbone Specifications

| Model | SA Layers | FFN Layers | MoE Layers | Att. Span | Embed Size | Params |
|---|---|---|---|---|---|---|
| *WikiText-103 Pretrain* | | | | | | |
| Switch-small | 3 | - | 3 | 256 | 128 | 70M |
| Switch-medium | 6 | - | 6 | 1024 | 352 | 216M |
| GLaM-small | 6 | 3 | 3 | 2048 | 144 | 79M |
| GLaM-medium | 12 | 6 | 6 | 2048 | 352 | 220M |
| *EnWik-8 Pretrain* | | | | | | |
| Switch | 8 | - | 8 | 2048 | 352 | 36M |

**Optimizer.** All experiments use Adam with a base learning rate of 0.0007. Small configurations use 3000 iterations of learning rate warmup while medium configurations use 4000 iterations.

**Pretrain Specification.** For WikiText-103 pretraining, small Switch backbones are trained for 40 epochs with a batch size of 96 and medium Switch backbones are trained for 80 epochs with a batch size of 48. Small GLaM backbones are trained for 60 epochs with a batch size of 48 and medium GLaM backbones are trained for 120 epochs with a batch size of 48. We use 0.01 auxiliary load balancing loss.

For EnWik-8 pretraining, both Switch and GLaM backbones are trained for 80 epochs with batch size 48. We use 0.01 auxiliary load balancing loss.

**Finetune Specification.** For SST2 and SST5 finetuning, we finetune for 5 epochs using Adam and a base learning rate of 0.001 without warmup and a batch size of 16. For B77 we finetune for 50 epochs using Adam and a base elarning rate of 0.00001 without warmup and a batch size of 16.

**Compute Resources.** All models are trained, evaluated, and finetuned on four NVIDIA A100 SXM4 40GB GPUs.

## C.2 IMAGE CLASSIFICATION

### C.2.1 DATASETS AND ATTACKS

**ImageNet-1K.** We use the full ImageNet dataset that contains 1.28M training images and 50K validation images. The model learns to predict the class of the input image among 1000 categories. We report the top-1 and top-5 accuracy on all experiments.

**ImageNet-A/O/R.** ImageNet-A (Hendrycks et al., 2021b) contains real-world adversarially filtered images that fool current ImageNet classifiers. A 200-class subset of the original ImageNet-1K's 1000 classes is selected so that errors among these 200 classes would be considered egregious, which cover most broad categories spanned by ImageNet-1K.

ImageNet-O (Hendrycks et al., 2021b) contains adversarially filtered examples for ImageNet out-of-distribution detectors. The dataset contains samples from ImageNet-22K but not from ImageNet1K, where samples that are wrongly classified as an ImageNet-1K class with high confidence by a ResNet-50 are selected.

Imagenet-R (Hendrycks et al., 2021a) contains various artistic renditions of object classes from the original ImageNet dataset, which is discouraged by the original ImageNet. ImageNet-R contains 30,000 image renditions for 200 ImageNet classes, where a subset of the ImageNet-1K classes is chosen.

**Adversarial Attacks.** We use produce corrupted ImageNet samples using white box attacks fast gradient sign method (FGSM) (Goodfellow et al., 2014) and projected gradient descent (PGD) (Madry et al., 2017), and black box simultaneous perturbation stochastic approximation (SPSA) (Uesato et al., 2018). FGSM and PGD use a perturbation budget of 1/255 while SPSA uses a perturbation budget 1. All attacks perturb under $l_\infty$ norm. PGD and uses 20 steps with step size of 0.15 and SPSA uses 20 iterations.

### C.2.2 MODEL, OPTIMIZER, & TRAIN SPECIFICATION

**Models.** Our results are based off of the Swin Transformer (Liu et al., 2021) architecture. This backbone uses 4 base layers of depth 2, 2, 18, and 2. The first two base layers each contain 2 self-attention layers and 2 feed-forward layers. The third base layer contains 18 self-attention layers with alternating feed-forward and MoE layers. The final base layer contains 2 self-attention layers with one feed-forward and one MoE layer. The embedding dimension is 96 and the heads per base layer are 3, 6, 12, and 24. We use 16 total experts and present results for both top-1 and top-2 expert routing. The total parameter count is 280M.

**Optimizer.** We use AdamW with a base learning rate of 1.25e-4, minimum learning rate of 1.25e-7, 0.1 weight decay and cosine scheduling.

**Train Specification.** We train for 60 epochs with a batch size of 128 and 0.1 auxiliary balancing loss.

**Compute Resources.** All models are trained and evaluated on four NVIDIA A100 SXM4 40GB GPUs.

### C.3 ADVERSARIAL ATTACK AT HIGHER PERTURBATION BUDGET

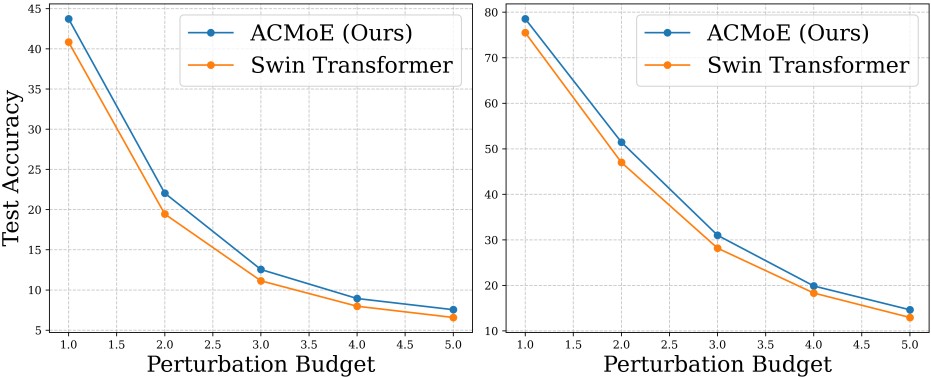

Figure 3: ACMoE and Swin Transformer under PGD attack at increasing perturbation budgets. ACMoE widens its performance gain over Swin at increasingly severe attacks in both top-1 test accuracy (**left**) and top-5 test accuracy (**right**), starting at approximately 7% improvement at 1/255 and ending at just over 10% at 5/255.

Figure 3 shows that for PGD perturbation budgets 1/255 through to 5/255, ACMoE widens its already substantive robust performance gain over Swin, with top-1 and top-5 test accuracy improvements increasing from 7% to approximately 10%.

### C.4 CLUSTER VISUALIZATION

We pass random ImageNet batches through Swin and ACMoE and plot the representations along with their assigned experts, using t-sne to represent the high dimensional data in 2 dimensions. The result is shown in Fig. 4, where we see Swin learns overlapping and indistinguishable expert clusters. ACMoE, on the other hand, performs better in learning the clusters, producing much clearer and better-distinguished clusters.

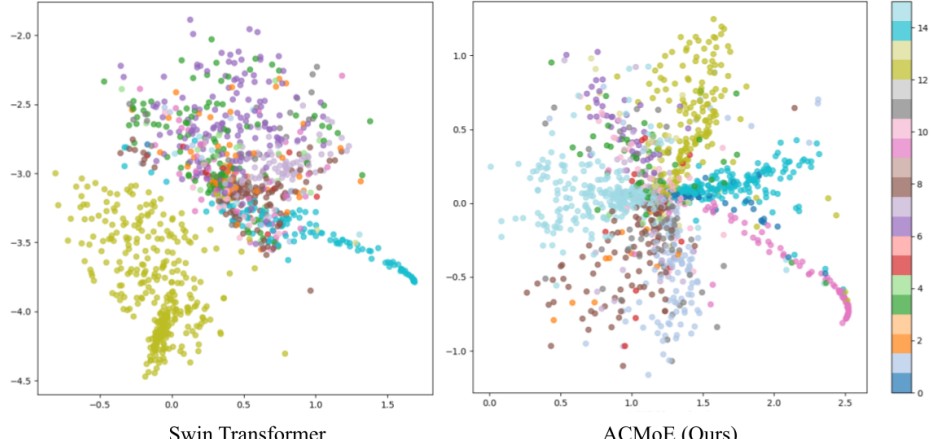

Figure 4: Cluster Visualization on ImageNet. Each token is represented as a point and colored by its assigned expert. **Left**: Swin identifies one cluster clearly (yellow/gold) but otherwise fails to distinguish remaining clusters **Right:** ACMoE learns better-defined expert clusters.

Table 9: Ablation on Measure of Spread in Switch Transformer (Fedus et al., 2022)

| Measure of Spread | Test PPL (↓) |
|---|---|
| Variance | 34.87 |
| MAD | **34.42** |

Table 10: Ablation on Layer Placement in Switch Transformer (Fedus et al., 2022)

| Layer Placement | Test PPL (↓) |
|---|---|
| Back Half | 34.95 |
| Alternating | 34.80 |
| Skip 1 | **34.42** |
| Full | 34.88 |

## C.5 ABLATION STUDIES

### C.5.1 MEASURES OF DISPERSION

We present in Tables 9 and 11 results for Switch-ACMoE and Swin-ACMoE when changing the measure of dispersion used in the AC routing transformation (Definition 1) from mean absolute deviation (MAD) to variance. We see mean absolute deviation outperforms variance as a measure of spread. This is an intuitive finding given that squared distances, as used in variance computations, are highly sensitive to outliers. Using mean absolute deviation as an alternative measure of spread reduces this issue and produces a more robust estimate of dispersion. We note that MAD is not the only robust measure of spread. We conjecture that taking interquartile range as an additionally robust measure of spread may produce good results in both clean and contaminated data. We, however, leave this interesting direction to future research as interquartile range poses implementation challenges as it requires designing concurrent linear scans over the expert clusters. MAD, by contrast, requires just two computations of the mean which is easily parallelizable using `torch.index_reduce()`.

### C.5.2 LAYER PLACEMENT

We consider the effect of layer placement in the Switch-medium configuration and in the Swin Transformer (see Sections C.1.2 and C.2.2 for the full model specifications). In particular, Switch is a 6 layer model and Swin is a 24 layer model. With regard to Swin, we focus on the deepest block of depth 18 to implement our ACMoE layers. This is due to the change in embedding size between base layers, meaning we are restricted to this base layer of depth 18. Note further that Swin only uses MoE layers in an alternating pattern with feed-forward networks between each MoE layer. For example, for Switch, a full ACMoE specification would mean placing ACMoE on layers 2,3,4,5,6. For Swin, a full specification means placing ACMoE on layers 4,6,8,10,12,14,16,18. To examine the effect of layer placement we consider the following models:

Table 11: Ablation on Measure of Spread in Swin Transformer

| Measure of Spread | Test Acc. | |
| --- | --- | --- |
| | Top 1 | Top 5 |
| *Swin-Top1* (Liu et al., 2021) | | |
| Variance | 75.06 | 92.49 |
| MAD | **75.39** | **92.56** |
| *Swin-Top2* (Liu et al., 2021) | | |
| Variance | 76.11 | 93.08 |
| MAD | **76.31** | **93.14** |

Table 12: Ablation on Layer Placement in Swin Transformer

| Layer Placement | Test Acc. | |
| --- | --- | --- |
| | Top 1 | Top 5 |
| *Swin-Top1* (Liu et al., 2021) | | |
| Back Half | 75.16 | 92.46 |
| Skip 2 | 75.34 | 92.42 |
| Skip 1 | 75.35 | 92.45 |
| Full | **75.39** | **92.56** |
| *Swin-Top2* (Liu et al., 2021) | | |
| Back Half | 76.16 | 93.02 |
| Skip 2 | 76.10 | 92.93 |
| Skip 1 | 76.29 | 92.98 |
| Full | **76.31** | **93.14** |

- *Alternating*: For Switch this means we place ACMoE on layers 2,4,6. For Swin this means we place ACMoE on layers 4,8,12,16.

- *Back Half*: For Switch this means we place ACMoE on just the last 3 layers of the network. For Swin this means we place ACMoE on just the last 5 layers of the network.

- *Skip 2*: For Swin this means we palce ACMoE on layers 8,10,12,14,16,18.

- *Skip 1*: For Switch this means we place ACMoE on layers 3,4,5,6. For Swin this means we place ACMoE on layers 6,8,10,12,14,16,18.

- *Full*: We place ACMoE on every possible layer.

We present in Table 10 results for Switch and Swin ACMoE models when changing the positions of the ACMoE layers throughout the network. The results agree with our expectation that, generally speaking, more ACMoE layers improve performance, but a in some circumstances a threshold is met at the point where ACMoE layers are used too early in the network such that the model has not been able to learn reasonably good approximations of the cluster membership of the tokens yet.

We find that in the Switch backbone, performance improves the more ACMoE layers we add, which agrees with our expectation that more ACMoE layers improve performance. However, we find that top performance is attained when allowing two standard MoE layers to go before the first ACMoE, as opposed to the minimum of 1 standard MoE layer. We conjecture this is because we need to give the model a few layers before the first ACMoE in order to learn decent representations such that we have good enough estimated cluster assignments for use in the ACMoE layer. Encouragingly, we find just one additional standard MoE layer is sufficient for the benefits of ACMoE to be obtained.

We find in Table 12 that with Swin, best performance is obtained using ACMoE on every possible layer, again agreeing with our expectation that more ACMoE layers improve performance. With Swin, however, we do not face any drop in performance from placing ACMoE too early in the network, and indeed we see *Full* attaining top performance. We conjecture that Swin does not encounter this issue since Swin uses four layers of feed forward networks before the first MoE layer, and so by the first MoE layer the representations are of reasonably good quality to produce good estimates of the cluster membership.

### C.5.3 RANDOM ABLATION

We show the efficacy of the adaptive clustering transformation $M$ (Definition 1) in our AC router at capturing meaningful feature-wise information by ablating it against an alternate $d \times d$ diagonal matrix made up of normal random variables with mean 1 and standard deviation 0.5 (where we clip any negative values to prevent negative weights). We present in Tables 13 and 14 results for language modeling (using Switch) and image classification (using Swin), which show fairly substantial drops in performance in both backbones. This offers evidence to the claim that our AC routing transformation is meaningfully weighting features to improve routing, and that performance gains

of our proposed method do not flow from a kind of implicit regularization of introducing noise into the router.

Table 13: Random Ablation in Switch (Fedus et al., 2022)

| Model | Test PPL (↓) |
|---|---|
| *Switch-Random* (Fedus et al., 2022) | 38.17 |
| Switch-ACMoE | **34.42** |

Table 14: Random Ablation in Swin (Liu et al., 2021)

| Model | Top 1 Acc. | Top 5 Acc. |
|---|---|---|
| *Swin-Random* | 74.22 | 91.87 |
| Swin-ACMoE | **76.31** | **93.14** |

## C.6 CLUSTER WEIGHT MIXING

The AC routing scheme estimates the cluster membership of each token based on its highest affinity cluster assigned in the previous layer. We could also further leverage the top-k structure of the MoE models by mixing the cluster-wise feature weights with weights corresponding to the affinities in the top-k routing. For example, if $h$ has affinity scores $\alpha$ and $1-\alpha$ to clusters $k$ and $k'$ respectively, then we could also obtain the required AC routing transformation for $h$ as $M_{k^*} = \alpha M_k + (1-\alpha)M_{k'}$. This approach therefore factors in the confidence with which we believe $h$ belongs to cluster $k$ or $k'$, and can be used for integrating ACMoE into higher expert granularity backbones (i.e higher top-k settings). Tables 15 and 16 show results for computing $M_{k^*}$ by mixing the top-affinity cluster weights (Mix 2) in Switch and GLaM with top-2 routing, versus our presented results which compute $M_{k^*}$ just based off of the highest affinity cluster (Mix 1). We see that GLaM-ACMoE benefits substantially from cluster weight mixing whereas Switch-ACMoE prefers just using its top affinity cluster weights. For consistency across models, we present in our main body the Mix 1 results, as GLaM-ACMoE already performs extremely strongly using Mix 1 and so we prefer to opt for the added performance gain in the Switch backbone.

Table 15: Results on Cluster Weight Mixing in Switch (Fedus et al., 2022)

| Clusters Mixed | Test PPL (↓) |
|---|---|
| Mix 2 | 34.66 |
| Mix 1 | **34.42** |

Table 16: Results on Cluster Weight Mixing in GLaM (Du et al., 2022)

| Clusters Mixed | Test PPL (↓) |
|---|---|
| Mix 2 | **35.29** |
| Mix 1 | 36.26 |

## C.7 ADAPTIVE CLUSTERING INTEGRATION INTO SOFT MIXTURE OF EXPERTS

We present here results for integrating ACMoE into SoftMoE (Puigcerver et al., 2023). To use ACMoE in the SoftMoe setting, which can be be understood as a top-E routing setting where all experts are active for every token, we compute $M_{k^*}$ using cluster weight mixing (Section C.6) over the top-8 highest affinity clusters. We present the performance of Soft-ACMoE on clean data, adversarially attacked data, and ImageNet-A/O/R in the following Tables 17 and 18.

Table 17: Test Accuracy on ImageNet corrupted PGD, FGSM, and SPSA using SoftMoE (Puigcerver et al., 2023) backbone

| Model | Clean Data | | PGD | | FGSM | | SPSA | |
|---|---|---|---|---|---|---|---|---|
| | Top 1 | Top 5 | Top 1 | Top 5 | Top 1 | Top 5 | Top 1 | Top 5 |
| *SoftMoE* (Puigcerver et al., 2023) | 72.86 | 90.92 | 45.29 | 78.91 | 56.95 | 85.60 | 66.59 | 88.70 |
| Soft-ACMoE (**Ours**) | **73.21** | **91.23** | **48.25** | **80.49** | **59.01** | **86.69** | **70.63** | **93.22** |

We see in Tables 17 and 18 the efficacy of ACMoE in the SoftMoE backbone, offering evidence of the adaptability of our framework into further MoE setups. In particular, the SoftMoE framework models a setting in which expert clusters are highly overlapping, as each token is soft assigned to all experts. Therefore, the performance gains shown in clean and contaminated data of Soft-ACMoE demonstrates that our AC router is well-suited to modeling such a clustering structure.

Table 18: Test Accuracy on Image Classification in Imagenet-A/O/R using SoftMoE (Puigcerver et al., 2023) backbone

| Model | Im-A Top-1 Acc. (↑) | Im-R Top-1 Acc. (↑) | Im-O AUPR (↑) |
|---|---|---|---|
| *SoftMoE* (Puigcerver et al., 2023) | 6.69 | 31.63 | 17.97 |
| Soft-ACMoE (**Ours**) | **6.93** | **32.18** | **18.35** |

## C.8 IMAGE CLASSIFICATION IN SWIN TRANSFORMER BASE CONFIGURATION

We further evaluate the performance ACMoE when scaling up model size in Table 19. We integrate ACMoE into the Base configuration of Swin (0.5B parameters) and evaluate on clean ImageNet-1K as well as under adversarial atacks.

Table 19: Test Accuracy on ImageNet corrupted PGD, FGSM, and SPSA using Swin Base (Liu et al., 2021) backbone

| Model | Clean Data | | PGD | | FGSM | | SPSA | |
|---|---|---|---|---|---|---|---|---|
| | Top 1 | Top 5 | Top 1 | Top 5 | Top 1 | Top 5 | Top 1 | Top 5 |
| *Swin-Base* (Liu et al., 2021) | 79.06 | 94.37 | 44.61 | 79.20 | 59.91 | 87.72 | 68.94 | 89.00 |
| Swin-ACMoE-Base (**Ours**) | **79.25** | **94.42** | **46.28** | **80.24** | **61.78** | 87.55 | **70.18** | **89.33** |

## C.9 ROUTER STABILITY

We present in Fig. 5 the routing stability of ACMoE, SMoE, XMoE, and StableMoE in the Switch backbone evaluated on WikiText-103. Routing instability computes over adjacent layers the proportion of tokens that are assigned to different experts across the two layers. Specifically, for $n$ tokens $[\boldsymbol{h}_1, \ldots, \boldsymbol{h}_n]$, we compute at layer $\ell$ the matrix $\boldsymbol{S}^\ell \in \mathbb{R}^{n \times n}$ such that $\boldsymbol{S}^\ell_{ij} = 1$ if the $i^{th}$ and $j^{th}$ tokens are assigned to the same expert in layer $\ell$ and is 0 otherwise. The router instability at layer $\ell$ can then be calculated as $r^\ell = \text{mean}(|\boldsymbol{S}^{\ell-1} - \boldsymbol{S}^\ell|)$. This metric therefore captures the degree to which tokens that are assigned to the same experts remain together through the model. A high $r^\ell$ indicates the router doesn't maintain consistent expert assignments, as tokens that it considers semantically similar at one layer it considers different at the next.

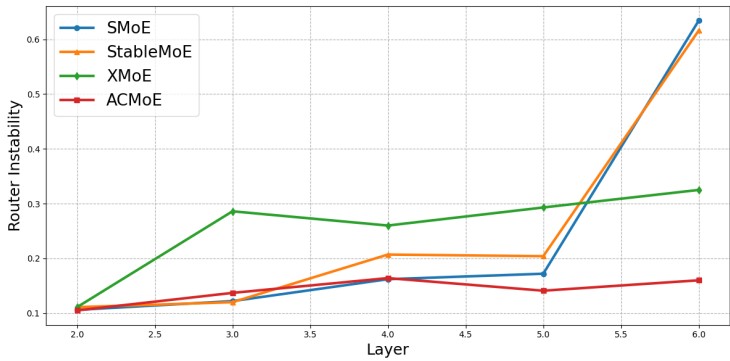

Figure 5: Router Instability of ACMoE, SMoE, XMoE, and StableMoE. ACMoE maintains consistent routing, while baseline routers more frequently change the expert assignments of tokens.

In Fig. 5, we see that baseline routers reach high levels of instability, where in the case of SMoE and StableMoE, at the last layer over 60% of tokens are assigned to a different expert. ACMoE, by contrast, maintains a more consistent, stable assignment through the model, with no more than 20% of tokens changing expert assignment across any layer.

## C.10 DYNAMIC ROUTING

We further test the compatibility of our Adaptive Clustering routing scheme in dynamic top-p routing. In this setting, rather than routing each token to its top-k highest affinity experts in each MoE

layer, we route each token to all experts that have affinity over a certain threshold $p$. This setting permits activating more or less experts for different tokens at different layers throughout the model, therefore dynamically assigning experts to tokens. We integrate our AC routing directly into this setting using the same setup as in Section 3, where the AC routing transformation is computed based on the estimated cluster membership of each token using the top affinity assignment of the previous layer. We present the results for Switch transformer on WikiText-103 language modeling in the following Table 20.

Table 20: Results on Top-$p$ Dynamic Routing in Switch Backbone (Fedus et al., 2022)

| Model | Test PPL ($\downarrow$) |
|---|---|
| *Fixed top-k routing* (Shazeer et al., 2017) | |
| *Switch-medium* (Fedus et al., 2022) | 35.48 |
| ACMoE-medium (**Ours**) | **34.42** |
| *Dynamic top-p routing (Guo et al., 2024)* | |
| *Switch-Fixed $p$* | 35.20 |
| Switch-ACMoE-Fixed $p$ (**Ours**) | **34.14** |
| *Switch-Learnable $p$* | 34.29 |
| Switch-ACMoE-Learnable $p$ (**Ours**) | **33.49** |

For fixed $p$, we set $p = 0.05$. For learnable $p$, we initialize the parameter to 0.05. We select this initialization as it reproduces approximately similar performance in the Switch backbone under default top-2 routing, thereby aiding direct comparison between fixed top-k and dynamic top-$p$ routing. We see in the dynamic routing setting, ACMoE maintains the same consistent improvement over the Switch baseline of roughly 1 full PPL. These results suggest ACMoE is well-suited to the dynamic routing setting.

## D    BROADER IMPACT

Our research offers benefits to Mixture-of-Expert (MoE) architectures in both clean and contaminated settings. In particular, our work offers socially beneficial outcomes with regard to defense against adversarial attack, which we hope can be used to protect important AI systems from malicious actors. Furthermore, as large language models, many of which are built on MoE backbones, continue to profligate and be used in important societal settings, we hope our improved robustness to data contamination can aid this promising technology to continue to grow and improve in realistic settings of noisy training and evaluation data. Our research also shows substantially faster convergence than comparative baselines. We believe this faster convergence can deliver significant social benefit in terms of reducing the energy requirements of large model training, thereby helping to ease the growing environmental burden of AI training runs. We recognize there will always be risk of misuse with AI systems, however we hope that our work can be used to enhance and protect socially beneficial AI while also decreasing the environmental impact of this technology. We furthermore hope that our research can spur others on to continue building on robust and efficient AI for social good.

