# OpenReview forum: "Tight Clusters Make Specialized Experts"
_ICLR.cc/2025/Conference — ICLR 2025 Poster_

### Official Review · Reviewer_2Ypo · 2024-10-30

**Soundness:** 3
**Presentation:** 3
**Contribution:** 3
**Rating:** 6
**Confidence:** 2

**Summary:**

The paper presents an Adaptive Clustering (AC) router for MoE models. The Ac router offers faster convergence, better robustness, and overall performance improvement.

**Strengths:**

The solution is theoretical based along with extensive evaluation.

**Weaknesses:**

- The paper targets MoE experts only, some of the notations and terms seem to be used specifically in the MoE domain.

**Questions:**

None

---

> ### Author Response · Authors · 2024-11-18
> **Rebuttal (1 / 1) : Weakness 1**
>
> **[Weakness 1. The paper targets MoE experts only, some of the notations and terms seem to be used specifically in the MoE domain.]**
>
> **Answer**
>
> Our work indeed targets MoE architectures, so the notation and terminology are typically taken from this literature. For future work, we do hope to extend our approach to further model families within deep learning. We would appreciate your further comment or suggestions on how to improve our notation and terms, which we would be happy to address during discussion.

---

> ### Author Response · Authors · 2024-11-21
> **Any Questions from Reviewer 2Ypo on Our Rebuttal?**
>
> We would like to thank the reviewer again for your thoughtful reviews and valuable feedback.
>
> We would appreciate it if you could let us know if our responses have addressed your concerns and whether you still have any other questions about our rebuttal.
>
> We would be happy to do any follow-up discussion or address any additional comments.

---

### Official Review · Reviewer_D6YM · 2024-10-31

**Soundness:** 3
**Presentation:** 3
**Contribution:** 3
**Rating:** 8
**Confidence:** 3

**Summary:**

the paper studies policies for selection of experts in sparse MoE models, and casts the problem as optimal clustering

the main contributions are 1) to derive closed form expression for the cluster weights. the formulation elegantly extends the classic topk MoE formulation using a diagonal matrix using inverse of cluster dispersion measure,  2) to theoretically bound the incorrect assignment probability and 3) prove faster convergence (though is not clear under which assumptions the prove holds

the paper presents experimental results on  text and image classification, mostly presenting results over baseline models

**Strengths:**

- the theoretical framework is solid, with clear explanations and remarks on the main body, and abundant material (only skimmed through) in the appendix concerning proofs  of the framework

- the paper is clearly written and motivated.

- the experimental section tackles two complementary domains and, whereas has less critical mass than the theoretical part, it is well executed

- extensive material in the appendix complement and completes the paper

**Weaknesses:**

# necessity of the framework

while the theoretical framework provides and the analysis in Tab 7 shows the method to only provide a rather modest additional complexity,  and the theoretical proofs shows it to be (in a sense) sufficient to solve the routing task, an open question is whether it is also necessary for the task.

otherwise stated, in the case of top-1/top-2 expert selection, an open question is whether it would be possible to significantly simplify the operations to be carried on (same way you do not need a full sort if you only need the maximum value of a vector) to attain similar performance ?

similarly, a more systematic comparison with related competitor work would better allow to appreciate the practical impact.

# structural difference of the solution

authors do a fair job in visually shown differences in the top experts (Fig 1), but that remains at an anecdotal level.  The load balancing analysis in sec 4.3 provides some complementary view, but is too aggregated (as it looks at measures of variation, but the baseline of perfect load balancing is clearly not a target as it likely would not be attainable).

as such, assessing and quantifying the /differences/ at individual expert level, or input level would provide a factual quantitative assessment of the number of times ACMoE expert selection differs from competing selections. (ie. instead of maps in Fig, providing a distribution of the times where top-k selection differed).

As in many cases the end-to-end performance improvement is slim (few percentage points) this intermediate assessment could allow to perceive the amount of top-k choices that were changed wrt the baselines models.


# deviation measures

it is unclear to what extent the proofs (and the practical performance) of the framework depends on subtle hyperparameter choices that are fixed and  not considered in the paper: e.g., measure of dispersion used in sec 3 is the mean absolute deviation, which is easier to compute but less robust than median absolute deviation (admittedly, this would go in the opposite direction as for simplification; but is unclear to what extent practical details as this one can have a performance impact in practice).

to have no impact on the selection, the top-k rank would need to be maintained across a family of measures of dispersion -- the impact of which is far from being clear after reading the paper. If the proof holds for families with given properties, the paper could be reinforced by making that clear.

**Questions:**

no further questions

---

> ### Author Response · Authors · 2024-11-18
> **Rebuttal (1 / 2) : Weaknesses 1 & 2**
>
> **[Weakness 1. Would it be possible to simplify the operations to attain similar performance? A more systematic comparison with related work would clarify the practical impact]**
>
> **Answer**
>
> We have added three additional comparisons with related work, which we present below. The first is using the SoftMoE backbone [1] (shown also in Table 17 of Appendix C.7), which shows that ACMoE is readily adaptable to problem settings in which expert clusters are broadly specialized and overlapping. The second and third are the Switch and GlaM backbones using the StableMoE [2] router (shown also in Table 3 of Section 4), where we see that ACMoE maintains consistent performance gains over StableMoE as well. Furthermore, we present in Figure 5 of Appendix C.9 additional empirical analysis for router stability, where we see ACMoE leads to substantially more stable routing against baseline routers.
>
> **Table 1**: Test Accuracy on ImageNet corrupted PGD, FGSM, and SPSA using SoftMoE [1] backbone
>
> | Model                 | Clean Data |      | PGD      |      | FGSM     |      | SPSA     |      |
> |-----------------------|------------|------|----------|------|----------|------|----------|------|
> |                       | Top 1      | Top 5| Top 1    | Top 5| Top 1    | Top 5| Top 1    | Top 5|
> | *SoftMoE* [1]         | 72.86      | 90.92| 45.29    | 78.91| 56.95    | 85.60| 66.59    | 88.70|
> | Soft-ACMoE (**Ours**) | **73.21**  | **91.23** | **48.25** | **80.49** | **59.01** | **86.69** | **70.63** | **93.22** |
>
> **Revised Table 3 of Section 4.2**: ACMoE in Switch and GlaM backbones, now showing additional StableMoE [2] results.
> | Router                                   | Test PPL (↓) |
> |------------------------------------------|--------------|
> | *Switch Transformer*|              |
> |                                          |              |
> | *SMoE-medium*     | 35.48        |
> | *XMoE-medium*          | 35.88        |
> | *StableMoE-medium*    | *35.33*      |
> | ACMoE-medium (**Ours**)                  | **34.42**    |
> |                                          |              |
> | *GLaM*                  |              |
> |                                          |              |
> | *SMoE-medium*     | 38.27        |
> | *XMoE-medium*          | 38.10        |
> | *StableMoE-medium*    | *38.04*      |
> | ACMoE-medium (**Ours**)                  | **36.26**    |
>
>
> Regarding the simplicity of the method, we agree there is always a possibility that we could perform our method with simplified computations and that this is a worthwhile avenue of research. Nonetheless, we present here two arguments for why we believe our method is already presented in a highly simplified form:
>
> 1. Computing MAD is a highly efficiently computed measure of dispersion, requiring just two computations of the mean, done in parallel. Computing variance requires additional squaring operations, and the calculations of interquartile range and median require cumbersome sorting operations.
> 2. We obtain our estimates of token cluster membership by simply retrieving the expert assignments from the previous layer, which avoids the need to explicitly cluster the tokens, which would typically require slow, iterative algorithms.
>
> [1]: Puigcerver et al. From Sparse to Soft Mixtures of Experts (ICLR 2023)
>
> [2]: Dai et al. StableMoE: Stable Routing Strategy for Mixture of Experts (ACL 2022)
>
> **[Weakness 2. Providing a figure on the times where top-k selection differed in ACMoE vs baselines could help clarify the proposed benefits of ACMoE with regard to load balance / distribution of expert activation]**
>
> **Answer**
>
> Thanks for your suggestion. We have added into Appendix C.9 of our revised manuscript Figure 5, showing the proportion of tokens for which the routing changed in ACMoE as compared with SMoE, XMoE, and StableMoE in the Switch backbone evaluated on WikiText-103. We see that XMoE maintains highly changeable routing throughout the model, while SMoE and StableMoE start off with consistent routing but by the final layer become unstable. ACMoE, by contrast, produces substantially more stable routing as compared with SMoE, XMoE and StableMoE, which complements the ability of experts to specialize by maintaining the routing of semantically similar tokens to the same experts.

---

> > ### Author Response · Authors · 2024-11-18
> > **Rebuttal (2 / 2) : Weakness 3**
> >
> > **[Weakness 3. It is unclear to what extent the performance of ACMoE depends on the measure of dispersion. The paper could be reinforced by considering performance over a family of dispersion measures]**
> >
> >
> > **Answer:**
> >
> > Thanks for your suggestion. We investigated different dispersion measures for our proposed AC Router and reported the results in Tables 9 and 11 in Appendix C.5.2 of our manuscript. We present those results here as well for viewing convenience. In particular, we see that using variance as the dispersion measure performs fairly similarly to MAD, which agrees with our expectation that the method should not be overly sensitive to the measure of dispersion. Nonetheless, MAD outperforms variance, which we hypothesize is due to MAD being a more robust metric, hence why we select it.
> >
> > We do note that interquartile range may also be an interesting measure of dispersion to try, but we do not test it as it would be prohibitively slow, since it would require sorting the tokens over all dimensions, per cluster.
> >
> > **Table 9 of Appendix C.5.1**: Ablation on measure of dispersion in Switch Transformer backbone
> > | Measure of Spread | Test PPL (↓) |
> > |-------------------|:------------:|
> > | Switch-ACMoE-Variance          |    34.87     |
> > | Switch-ACMoE-MAD **(Ours)**              |  **34.42**   |
> >
> > **Table 11 of Appendix C.5.1**: Ablation on measure of dispersion in Swin Transformer backbone
> > | Measure of Spread | Top 1  | Top 5  |
> > |-------------------|--------|--------|
> > | Swin-ACMoE-Top 1-Variance          | 75.06  | 92.49  |
> > | Swin-ACMoE-Top 1-MAD **(Ours)**              | **75.39** | **92.56** |
> > |                   |        |        |
> > | Swin-ACMoE-Top 2-Variance          | 76.11  | 93.08  |
> > | Swin-ACMoE-Top 2-MAD **(Ours)**            | **76.31** | **93.14** |

---

> ### Author Response · Authors · 2024-11-21
> **Any Questions from Reviewer D6YM on Our Rebuttal?**
>
> We would like to thank the reviewer again for your thoughtful reviews and valuable feedback.
>
> We would appreciate it if you could let us know if our responses have addressed your concerns and whether you still have any other questions about our rebuttal.
>
> We would be happy to do any follow-up discussion or address any additional comments.

---

> > ### Comment · Reviewer_D6YM · 2024-11-26
> >
> > thanks for the detailed answers.  I was already positive with this paper, and my comments were minor, so the answers confirm the current rating

---

> > > ### Author Response · Authors · 2024-11-27
> > > **Thanks for your endorsement!**
> > >
> > > Thanks for your response, and we appreciate your endorsement.

---

### Official Review · Reviewer_LDye · 2024-11-03

**Soundness:** 4
**Presentation:** 3
**Contribution:** 3
**Rating:** 8
**Confidence:** 3

**Summary:**

This paper proposes a novel routing mechanism called Adaptive Clustering (AC) router for Mixture-of-Experts (MoE) architectures. The key idea is to compute token-expert assignments in an adaptively transformed space that better reveals latent clusters in the data. The transformation is derived from a feature-weighted clustering optimization perspective, where features that promote tight clustering for each expert are upweighted. The authors demonstrate both theoretical and empirical advantages of their method, showing improved convergence speed, robustness to data contamination, and overall performance across language modeling and image classification tasks.

**Strengths:**

1. The paper introduces a novel perspective by viewing the MoE routing mechanism through the lens of feature-weighted clustering optimization. This theoretical framing allows for the derivation of optimal feature weights that improve cluster separability.

2. The authors provide rigorous theoretical analysis that support their claims on the improved robustness and convergence speed of their method.

3. Extensive experiments on large-scale datasets for both language modeling and image classification demonstrate the effectiveness of the AC router.

4. The proposed method requires no additional learnable parameters and introduces negligible computational overhead. And it can be integrated into any existing MoE architecture

**Weaknesses:**

1. The AC router relies on expert assignments from the previous layer to compute the adaptive transformation. This dependence may limit its applicability in scenarios where embedding sizes change between layers or in networks without consistent layer structures.

2. The effectiveness of the AC router is closely tied to the quality of expert assignments in earlier layers. Since the method relies on routing decisions from prior layers, it may be less effective if the early layers have not yet learned meaningful or well-separated cluster structures. In practice, early layers in deep networks often learn basic features that may not be semantically distinct enough to form tight clusters. As a result, the AC router may struggle to make optimal routing decisions in later layers if the cluster assignments in the initial layers are poor.

3. The method assumes that the input data naturally clusters in the feature space. While this assumption holds in many cases (e.g., language modeling and image classification), it might not generalize to tasks where the input data is more uniformly distributed or lacks clear clustering patterns.

**Questions:**

1. The choice of mean absolute deviation (MAD) as the measure of dispersion is justified, but the paper does not deeply explore the impact of other potential measures, such as variance or interquartile range, on performance. Have you considered or tested other measures of dispersion besides MAD? How sensitive is the method to the choice of dispersion measure, and could alternative measures potentially improve performance or robustness?

2. How does the AC router handle situations where the assignments from previous layers are noisy or not well-defined? Is there a way to initialize or adjust the method to be effective in early layers or when prior assignments are unreliable?

3. Since the method requires consistent embedding dimensions between layers, how can it be adapted for architectures where the embedding size changes, such as in some convolutional neural networks or transformers with variable dimensions?

4. The theoretical analysis relies on Gaussian mixture models. How does the AC router perform when the data clusters have non-Gaussian distributions or are not well-separated?

---

> ### Author Response · Authors · 2024-11-18
> **Rebuttal (1/ 6) : Weaknesses 1 & 2**
>
> **[Weakness 1. The AC router relies on expert assignments from the previous layer, which may limit its applicability in scenarios where embedding sizes change between layers]**
>
> **Answer**
>
> In the majority of contemporary deep MoE architectures, the embedding size is typically constant throughout the entire model, and examples of changing embedding sizes are more the exception than the rule. For example, in Switch [1], GLaM [2], SoftMoE [3], and VMoE [4], the embedding size remains the same throughout. Swin [5] is, so far, the only transformer-MoE model we've encountered that features a changing embedding size, but still maintains the same embedding size for 18 of its 22 total layers, and so there remains ample opportunity for applying ACMoE.
>
> We note further that many standard-practice architectural features require the same assumption of constant embedding size, such as the residual connection. So in this sense, our requirement of constant embedding size at adjacent MoE layers is no more restrictive than what is required for commonplace design choices.
>
> [1] Fedus et al. Switch Transformers: Scaling to Trillion Parameter Models with Simple and Efficient Sparsity (JMLR 2022)
> [2] Du et al. GLaM: Efficient Scaling of Language Models with Mixture-of-Experts (ICML 2022)
> [3] Puigcerver et al. From Sparse to Soft Mixtures of Experts (ICLR 2023)
> [4] Riquelme et al. Scaling Vision with Sparse Mixture of Experts (NeurIPS 2021)
> [5] Liu et al. Swin Transformer: Hierarchical Vision Transformer using Shifted Windows (ICCV 2021)
>
>
>
> **[Weakness 2: The effectiveness of the AC router is tied to the quality of expert assignments in earlier layers. Since the method relies on routing decisions from prior layers, it may be less effective if the early layers have not yet learned meaningful cluster structures. As a result, the AC router may struggle to make optimal routing decisions in later layers if the cluster assignments in the initial layers are poor.]**
>
> **Answer**
>
> We agree with the reviewer and indeed initially shared this concern when first desigining the method. Nonetheless, encouragingly, we actually find that ACMoE not only outperforms baselines when only applied in later layers, but attains best performance when applied in early layers as well. This offers evidence that even at early layers before the experts have learnt fine-grained or well-separated structures, there is still enough information in the cluster assignments that we can leverage to meaningfully apply our AC routing transformation and obtain the proposed benefits. We refer the reviewer to Appendix C.5.2 for the results, and display them here as well for convenience. Our empirical results suggest that in Swin the best performance is attained with the AC router on every possible layer. For Switch, we see a small performance bump by skipping the first layer as opposed to *Full*, but that *Skip 1* still outperforms *Back Half*, indicating that earlier placement of the AC router is still beneficial.
>
> **Swin-ACMoE Ablation Study on AC router layer placement**
>
> | Layer Placement   | Top 1  | Top 5  |
> |-------------------|--------|--------|
> | **Swin-ACMoE-Top1**|        |        |
> | Back Half         | 75.16  | 92.46  |
> | Skip 2            | 75.34  | 92.42  |
> | Skip 1            | 75.35  | 92.45  |
> | Full              | **75.39** | **92.56** |
> |                   |        |        |
> | **Swin-ACMoE-Top2**  |        |        |
> | Back Half         | 76.16  | 93.02  |
> | Skip 2            | 76.10  | 92.93  |
> | Skip 1            | 76.29  | 92.98  |
> | Full              | **76.31** | **93.14** |
>
>
> **Switch-ACMoE Ablation Study on AC router layer placement**
> | Layer Placement   | Test PPL (↓)  |
> |-------------------|---------------|
> | Back Half         | 34.95         |
> | Alternating       | 34.80         |
> | Skip 1            | **34.42**     |
> | Full              | 34.88         |
>
>
> The names are as follows:
>
> *Full*: AC router on every layer
>
> *Alternating*: AC router on alternating layers
>
> *Skip 1*: AC router on every layer except for the first possible layer
>
> *Skip 2*: AC router on every layer except for the first  two possible layers
>
> *Back Half*: AC router only on the back half of the total layers

---

> ### Author Response · Authors · 2024-11-18
> **Rebuttal (2 / 6) : Weakness 3 (1/2)**
>
> **[Weakness 3. The method assumes that the input data naturally clusters in the feature space. While this assumption holds in many cases (e.g., language modeling and image classification), it might not generalize to tasks where the input data is more uniformly distributed or lacks clear clustering patterns.]**
>
> **Answer**
>
> Thanks for your comments. Below we address your concerns about: 1) the setting to which our approach is well-tailored, and 2) the performance of our model when latent clusters are indistinct, and we address them each in turn.
>
> **The setting to which the approach is well-tailored.** Though we do expect our approach to work well in settings with well-structured clusters, our method is actually more so motivated by settings in which clusters are not easily identified. From the perspective of cluster analysis, these are challenging problem settings in which classical clustering algorithms typically fail to discover the clustering structure of the data in the untransformed feature space [6]. This is why our feature-weighted clustering optimization setup for the proposed AC router in Eqns. 4 and 5 explicitly use different weights for each cluster so that we permit the possibility that clusters depend on differing, possibly disjoint, sets of features. We also validate our AC router on large-scale, natural datasets such as WikiText-103 and ImageNet, where the latent clusters in the input distribution may not be easily discoverable. As shown in Table 1, 2, 3, 4, and 5 in the main text, our AC router improves the accuracy and robustness of the baseline model on these benchmarks.
>
> We would also like to clarify that we do not choose the features along which experts cluster. Instead, we *learn* features from the data, and use them to transform the space in which routing takes place to improve token-expert matching.

---

> ### Author Response · Authors · 2024-11-18
> **Rebuttal (3 / 6) : Weakness 3 (2/2)**
>
> **Performance of our model when clusters are indistinct.** In addition to the experiments on WikiText-103 and ImageNet mentioned above, to more explicitly justify the performance of our model in this setting, we integrate our AC router into the SoftMoE [7], in which each token is soft-assigned to every expert, thereby modeling more broadly specialized, overlapping expert clusters. We also perform additional studies on our method using cluster weight mixing, which models this same scenario of overlapping clusters, in the Switch and GlaM backbones. Please see the additional results in the global response, where we see that our framework adapts well to this setting and continues to deliver strong performance gains. We paste the results here as well for viewing convenience:
>
> **Table 1**: Test Accuracy on ImageNet corrupted PGD, FGSM, and SPSA using SoftMoE [7] backbone
>
> | Model                 | Clean Data |      | PGD      |      | FGSM     |      | SPSA     |      |
> |-----------------------|------------|------|----------|------|----------|------|----------|------|
> |                       | Top 1      | Top 5| Top 1    | Top 5| Top 1    | Top 5| Top 1    | Top 5|
> | *SoftMoE* [7]         | 72.86      | 90.92| 45.29    | 78.91| 56.95    | 85.60| 66.59    | 88.70|
> | Soft-ACMoE (**Ours**) | **73.21**  | **91.23** | **48.25** | **80.49** | **59.01** | **86.69** | **70.63** | **93.22** |
>
> **Table 2**: Test Accuracy on ImageNet corrupted PGD, FGSM, and SPSA using Swin Base [8] backbone and cluster mixing over top-2 highest affinity experts
>
> | Model                   | Clean Data |      | PGD      |      | FGSM     |      | SPSA     |      |
> |-------------------------|------------|------|----------|------|----------|------|----------|------|
> |                         | Top 1      | Top 5| Top 1    | Top 5| Top 1    | Top 5| Top 1    | Top 5|
> | *Swin-Base* [8]         | 79.06      | 94.37| 44.61    | 79.20| 59.91    | 87.72| 68.94    | 89.00|
> | Swin-ACMoE-Mix 2-Base (**Ours**) | **79.25**  | **94.42** | **46.28** | **80.24** | **61.78** | **87.55** | **70.28** | **89.38** |
>
> **Table 3**: ACMoE with cluster weight mixing in Switch [9] and GLaM [10]
>
> | Clusters Mixed          | Test PPL (↓) |
> |-------------------------|--------------|
> | *Switch Transformer* [9]| 35.48        |
> | Switch-ACMoE-Mix 2      | 34.66        |
> | Switch-ACMoE-Mix 1  **(Ours)**       | **34.42**    |
> |                         |              |
> | *GLaM* [10]              | 38.27        |
> | GLaM-ACMoE-Mix 2        | **35.29**    |
> | GLaM-ACMoE-Mix 1  **(Ours)**       | 36.26        |
>
> Finally, we would like to make a remark about scenarios where clusters are indistinct. The existence of latent clusters within the input distribution is a fundamental and motivating assumption underlying the MoE framework [11,12], as the framework is largely built upon this premise. Consequently, we argue that if, in the extreme, an input distribution contains completely indistinguishable and overlapping clusters, such a situation may render the MoE framework as a whole less suitable. In this context, our assumption of identifiable clusters in the input distribution is no more restrictive than the foundational assumption of the MoE framework itself. Our model is specifically designed to address scenarios where clusters are challenging to identify due to their varying dependencies on subsets of features. Furthermore, as demonstrated by our additional results involving cluster weight mixing and SoftMoE, our approach continues to perform effectively in settings where expert clusters are highly overlapping.
>
> [6] Witten & Tibshirani, A framework for feature selection in clustering (Journal of the American Statistical Association, 2010)
>
> [7] Puigcerver et al. From Sparse to Soft Mixtures of Experts (ICLR 2023)
>
> [8] Liu et al. Swin Transformer: Hierarchical Vision Transformer using Shifted Windows (ICCV 2021)
>
> [9] Fedus et al. Switch Transformers: Scaling to Trillion Parameter Models with Simple and Efficient Sparsity (JMLR 2022)
>
> [10] Du et al. GLaM: Efficient Scaling of Language Models with Mixture-of-Experts (ICML 2022)
>
> [11] Robert Jacobs, Michael I Jordan, Steven J Nowlan, and Geoffrey E Hinton. Adaptive mixtures of local experts (Neural computation 1991)
>
> [12] David Eigen, Marc'Aurelio Ranzato, Ilya Sutskever. Learning Factored Representations in a Deep Mixture of Experts (ICLR 2014)

---

> ### Author Response · Authors · 2024-11-18
> **Rebuttal (4 / 6) : Questions 1 & 2**
>
> **[Question 1. Have you considered or tested other measures of dispersion besides MAD? How sensitive is the method to the choice of dispersion measure, and could alternative measures potentially improve performance or robustness?]**
>
> **Answer**
>
> Thanks for your suggestion. We investigated different dispersion measures for our proposed AC Router and reported the results in Tables 9 and 11 in Appendix C.5.2 of our manuscript. We present those results here as well for viewing convenience. In particular, we see that using variance as the dispersion measure performs fairly similarly to MAD, which agrees with our expectation that the method should not be overly sensitive to the measure of dispersion. Nonetheless, MAD outperforms variance, which we hypothesize is due to MAD being a more robust metric, hence why we select it.
>
> We do note that interquartile range may also be an interesting measure of dispersion to try, but we do not test it as it would be prohibitively slow, since it would require sorting the tokens over all dimensions, per cluster.
>
> **Table 9 of Appendix C.5.1. Ablation on measure of dispersion in Switch Transformer backbone**
> | Measure of Spread | Test PPL (↓) |
> |-------------------|:------------:|
> | Switch-ACMoE-Variance          |    34.87     |
> | Switch-ACMoE-MAD **(Ours)**              |  **34.42**   |
>
> **Table 11 of Appendix C.5.1. Ablation on measure of dispersion in Swin Transformer backbone**
> | Measure of Spread | Top 1  | Top 5  |
> |-------------------|--------|--------|
> | Swin-ACMoE-Top 1-Variance          | 75.06  | 92.49  |
> | Swin-ACMoE-Top 1-MAD **(Ours)**              | **75.39** | **92.56** |
> |                   |        |        |
> | Swin-ACMoE-Top 2-Variance          | 76.11  | 93.08  |
> | Swin-ACMoE-Top 2-MAD **(Ours)**            | **76.31** | **93.14** |
>
>
>
> **[Question 2. How does the AC router handle situations where the assignments from previous layers are noisy or not well-defined? Is there a way to initialize or adjust the method to be effective in early layers or when prior assignments are unreliable?]**
>
> **Answer**
>
> The AC router can indeed be straightforwardly applied in situations when prior assignments are noisy. Below, we demonstrate the efficacy of our AC router using cluster weight mixing, in which we soften our estimated cluster assignments by modeling the confidence with which we believe a token belongs to any cluster in the top-k routing assignment. Furthermore, as shown by the results using SoftMoE, our ACMoE performs well in a setting where all experts are active for each token, representing a setting in which expert clusters are highly overlapping. In all cases, we see ACMoE is readily adaptable to these settings of noisy or ill-defined clusters and is able to continue delivering the proposed performance gains.
>
> **Table 1**: Test Accuracy on ImageNet corrupted PGD, FGSM, and SPSA using SoftMoE [1] backbone
>
> | Model                 | Clean Data |      | PGD      |      | FGSM     |      | SPSA     |      |
> |-----------------------|------------|------|----------|------|----------|------|----------|------|
> |                       | Top 1      | Top 5| Top 1    | Top 5| Top 1    | Top 5| Top 1    | Top 5|
> | *SoftMoE*         | 72.86      | 90.92| 45.29    | 78.91| 56.95    | 85.60| 66.59    | 88.70|
> | Soft-ACMoE (**Ours**) | **73.21**  | **91.23** | **48.25** | **80.49** | **59.01** | **86.69** | **70.63** | **93.22** |
>
> **Table 2**: Test Accuracy on ImageNet corrupted PGD, FGSM, and SPSA using Swin Base backbone and cluster mixing over top-2 highest affinity experts
>
> | Model                   | Clean Data |      | PGD      |      | FGSM     |      | SPSA     |      |
> |-------------------------|------------|------|----------|------|----------|------|----------|------|
> |                         | Top 1      | Top 5| Top 1    | Top 5| Top 1    | Top 5| Top 1    | Top 5|
> | *Swin-Base*         | 79.06      | 94.37| 44.61    | 79.20| 59.91    | 87.72| 68.94    | 89.00|
> | Swin-ACMoE-Mix 2-Base (**Ours**) | **79.25**  | **94.42** | **46.28** | **80.24** | **61.78** | **87.55** | **70.28** | **89.38** |
>
> **Table 3**: ACMoE with cluster weight mixing in Switch and GLaM
>
> | Clusters Mixed          | Test PPL (↓) |
> |-------------------------|--------------|
> | *Switch Transformer*| 35.48        |
> | Switch-ACMoE-Mix 2      | 34.66        |
> | Switch-ACMoE-Mix 1  **(Ours)**       | **34.42**    |
> |                         |              |
> | *GLaM*              | 38.27        |
> | GLaM-ACMoE-Mix 2        | **35.29**    |
> | GLaM-ACMoE-Mix 1  **(Ours)**       | 36.26        |

---

> ### Author Response · Authors · 2024-11-18
> **Rebuttal (5 / 6) : Question 3**
>
> **[Question 3. Since the method requires consistent embedding dimensions between layers, how can it be adapted for architectures where the embedding size changes?]**
>
> **Answer**
>
> We break our answer to this question down into two parts. First, we address the prevalence of consistent embedding size between layers, and second, we provide two future research directions for how to nonetheless apply our framework in this situation.
>
> First, in the majority of contemporary deep MoE architectures, the embedding size is typically constant throughout the entire model, and examples of changing embedding sizes are more the exception than the rule. For example, in Switch, GLaM, SoftMoE, and VMoE, the embedding size remains the same throughout. Swin is, so far, the only transformer-MoE model we've encountered that features a changing embedding size, but still maintains the same embedding size for 18 of its 22 total layers, and so there remains ample opportunity for applying ACMoE.
>
> We note further that many standard-practice architectural features require the same assumption of constant embedding size, such as the residual connection. So in this sense, our requirement of constant embedding size at adjacent MoE layers is no more restrictive than what is required for commonplace design choices.
>
> As to the question of how might the AC router by used be  in situations where the embedding size changes at two adjacent layers, we present two possible future research directions for applying the AC router framework without using previous layer assignments as estimates of cluster membership:
>
> a) One could estimate the clustering structure of the input tokens without reliance on previous layer expert assignments by applying a few steps of a clustering algorithm to the tokens before sending them into the router. A straightforward choice would therefore be to use k-means clustering on the input tokens. To reduce expense of this approach, one could try just one or two steps of k-means.
>
> b) In vision-only settings, one could also try applying an image segmenter before the router to segment the tokens into semantically similar groupings.
>
> We do think it worth noting, however, that we don't see reliance on previous layers as an overly burdensome or problematic scheme for obtaining the cluster assignments. Indeed, this method is efficient, simple to implement, works well empirically, and is justified by the mild assumption that previous layer assignments are good estimates of token cluster membership. When compared with alternate ideas for obtaining the estimates of the cluster membership of the tokens (for example the two above ideas of k-means and image segmenters), we note that our proposed method would be much more efficient.

---

> > ### Author Response · Authors · 2024-11-18
> > **Rebuttal (6 / 6) : Question 4**
> >
> > **[Question 4. The theoretical analysis relies on Gaussian mixture models. How does the AC router perform when the data clusters have non-Gaussian distributions or are not well-separated?]**
> >
> > **Answer**
> >
> > While the Gaussian mixture model (GMM) assumption is a fair concern, we argue below that it does not affect the validity of our theoretical propositions nor significantly impact the applicability of our model.
> >
> > Theoretically, a GMM is a universal approximator of densities, in the sense that any smooth density can be approximated with any precision by a GMM with enough components, while linear combinations of translations of a single canonical Gaussian are also shown to be dense in $L^2(\mathbb R)$ [13, 14]. Existing universality theorems tailored for the MoE neural network architecture, such as in [15], further justify the broad applicability of this theoretical assumption fundamental to all MoE architectures.
> >
> > Empirically, our experiments are conducted on real-world data, such as ImageNet and WikiText-103, rather than data generated from GMMs, which strengthen the justification for using the AC router in scenarios that extend beyond the GMM framework. These results demonstrate the practical versatility of the proposed approach in handling complex, non-GMM distributions. Futhermore, please see the additional results in Tables 1 and 3 of the global response where we show how our framework offers improvements in settings where overlapping and highly fine-grained clusters are explicitly modeled, such as in the SoftMoE backbone. We see that our framework continues to deliver strong performance gains in this setting.
> >
> > We note as well that while non-GMM theoretical results could be valuable, they may come with trade-offs in practical interpretability and increased complexity. In particular, we identify the following challenges:
> > - **Lack of Parametric Structure**: Without GMMs, the density functions lack a simple parametric form, making it difficult to analyze and model complex behaviors.
> > - **Increased Analytical Complexity**: Proving convergence, error bounds, and identifiability for arbitrary distributions often leads to undesirable abstractions in the theoretical results.
> > - **Limited Empirical Verifiability**: Abovementioned abstractions outside the GMM framework makes it harder to empirically validate theoretical findings in practical settings.
> >
> > [13]: Goodfellow, I., Bengio, Y., & Courville, A. (2016). Deep learning., p. 65
> >
> > [14]: Calcaterra, C., & Boldt, A. (2008). Approximating with Gaussians. arXiv preprint arXiv:0805.3795.
> >
> > [15]: Nguyen, H. D., Lloyd-Jones, L. R., & McLachlan, G. J. (2016). A universal approximation theorem for mixture-of-experts models. Neural Computation, 28(12), p. 2585–2593.

---

> ### Author Response · Authors · 2024-11-21
> **Any Questions from Reviewer LDye on Our Rebuttal?**
>
> We would like to thank the reviewer again for your thoughtful reviews and valuable feedback.
>
> We would appreciate it if you could let us know if our responses have addressed your concerns and whether you still have any other questions about our rebuttal.
>
> We would be happy to do any follow-up discussion or address any additional comments.

---

> > ### Comment · Reviewer_LDye · 2024-11-23
> >
> > Thank you for your detailed responses and clarifications. I appreciate the effort you put into addressing my concerns. Based on the revisions and responses, I am raising my score to 8.

---

> > > ### Author Response · Authors · 2024-11-23
> > > **Thanks for your endorsement!**
> > >
> > > Thanks for your response, and we appreciate your endorsement.

---

### Official Review · Reviewer_UqYP · 2024-11-06

**Soundness:** 3
**Presentation:** 3
**Contribution:** 3
**Rating:** 6
**Confidence:** 3

**Summary:**

This paper proposes an Adaptive Clustering (AC) router for Sparse Mixture-of-Experts (MoE) architectures to improve routing efficiency. It focuses on optimizing the matching of tokens to experts by adaptively transforming the input feature space. This method promotes faster convergence, increased robustness against data contamination, and improved overall performance. The authors demonstrate the AC router's effectiveness in large-scale language and image tasks, outperforming baseline routers in robustness and efficiency without added learnable parameters.

**Strengths:**

1.	The proposed router uses a unique, feature-weighted approach for adaptive clustering, enabling more efficient specialization in MoE models. It achieves improvements without extra learnable parameters or added computation cost, making it ideal for scalable applications.

2.	The paper provides robust theoretical proof supporting the method's faster convergence and increased robustness

3.	The experiments cover various large-scale tasks, such as WikiText-103 and ImageNet, and include backbones like Switch Transformer, GLaM, and Swin Transformers, evaluating performance under both clean and corrupted data conditions.

**Weaknesses:**

1.	The approach seems to tailor to settings with well-structured clusters, which may limit performance on datasets or tasks where latent clusters are less distinct or not aligned with chosen features.

2.	The AC router relies on prior layers for initial expert assignments, potentially constraining flexibility and requiring consistent embedding sizes across layers.

3.	Although tested on large-scale datasets, further application to more dynamic datasets could strengthen claims of robustness and adaptability.

**Questions:**

1.	How does the AC router handle scenarios with overlapping or dynamically evolving clusters where expert specialization may be less clear?

2.	What mechanisms could be integrated to enhance adaptability in routing without relying on previous layer assignments?

---

> ### Author Response · Authors · 2024-11-18
> **Rebuttal (1/3) : Weakness 1 and Question 1 (1/2)**
>
> **[Weakness 1 & Question 1:  The approach seems to tailor to settings with well-structured clusters, which may limit performance where latent clusters are less distinct or not aligned with chosen features. How does the AC router handle overlapping clusters where expert specialization may be less clear?]**
>
> **Answer**
>
> Thanks for your comments. Below we address your concerns about: 1) the setting to which our approach is well-tailored, and 2) the performance of our model when latent clusters are indistinct, and we address them each in turn.
>
> **The setting to which the approach is well-tailored.** Though we do expect our approach to work well in settings with well-structured clusters, our method is actually more so motivated by settings in which clusters are not easily identified. From the perspective of cluster analysis, these are challenging problem settings in which classical clustering algorithms typically fail to discover the clustering structure of the data in the untransformed feature space [1]. This is why our feature-weighted clustering optimization setup for the proposed AC router in Eqns. 4 and 5 explicitly use different weights for each cluster so that we permit the possibility that clusters depend on differing, possibly disjoint, sets of features. We also validate our AC router on large-scale, natural datasets such as WikiText-103 and ImageNet, where the latent clusters in the input distribution may not be easily discoverable. As shown in Table 1, 2, 3, 4, and 5 in the main text, our AC router improves the accuracy and robustness of the baseline model on these benchmarks.
>
> We would also like to clarify that we do not choose the features along which experts cluster. Instead, we *learn* features from the data, and use them to transform the space in which routing takes place to improve token-expert matching.

---

> ### Author Response · Authors · 2024-11-18
> **Rebuttal (2/3) : Weakness 1 and Question 1 (2/2)**
>
> **Performance of our model when clusters are indistinct.** In addition to the experiments on WikiText-103 and ImageNet mentioned above, to more explicitly justify the performance of our model in this setting, we integrate our AC router into the SoftMoE, in which each token is soft-assigned to every expert, thereby modeling more broadly specialized, overlapping expert clusters. We also perform additional studies on our method using cluster weight mixing, which models this same scenario of overlapping clusters, in the Switch and GlaM backbones. Please see the additional results in the global response, where we see that our framework adapts well to this setting and continues to deliver strong performance gains. We paste the results here as well for viewing convenience:
>
> **Table 1**: Test Accuracy on ImageNet corrupted PGD, FGSM, and SPSA using SoftMoE backbone
>
> | Model                 | Clean Data |      | PGD      |      | FGSM     |      | SPSA     |      |
> |-----------------------|------------|------|----------|------|----------|------|----------|------|
> |                       | Top 1      | Top 5| Top 1    | Top 5| Top 1    | Top 5| Top 1    | Top 5|
> | *SoftMoE*         | 72.86      | 90.92| 45.29    | 78.91| 56.95    | 85.60| 66.59    | 88.70|
> | Soft-ACMoE (**Ours**) | **73.21**  | **91.23** | **48.25** | **80.49** | **59.01** | **86.69** | **70.63** | **93.22** |
>
> **Table 2**: Test Accuracy on ImageNet corrupted PGD, FGSM, and SPSA using Swin Base backbone and cluster mixing over top-2 highest affinity experts
>
> | Model                   | Clean Data |      | PGD      |      | FGSM     |      | SPSA     |      |
> |-------------------------|------------|------|----------|------|----------|------|----------|------|
> |                         | Top 1      | Top 5| Top 1    | Top 5| Top 1    | Top 5| Top 1    | Top 5|
> | *Swin-Base*         | 79.06      | 94.37| 44.61    | 79.20| 59.91    | 87.72| 68.94    | 89.00|
> | Swin-ACMoE-Mix 2-Base (**Ours**) | **79.25**  | **94.42** | **46.28** | **80.24** | **61.78** | **87.55** | **70.28** | **89.38** |
>
> **Table 3**: ACMoE with cluster weight mixing in Switch and GLaM
>
> | Clusters Mixed          | Test PPL (↓) |
> |-------------------------|--------------|
> | *Switch Transformer*| 35.48        |
> | Switch-ACMoE-Mix 2      | 34.66        |
> | Switch-ACMoE-Mix 1  **(Ours)**       | **34.42**    |
> |                         |              |
> | *GLaM*              | 38.27        |
> | GLaM-ACMoE-Mix 2        | **35.29**    |
> | GLaM-ACMoE-Mix 1  **(Ours)**       | 36.26        |
>
> Finally, we would like to make a remark about scenarios where clusters are indistinct. The existence of latent clusters within the input distribution is a fundamental and motivating assumption underlying the MoE framework [2,3], as the framework is largely built upon this premise. Consequently, we argue that if, in the extreme, an input distribution contains completely indistinguishable and overlapping clusters, such a situation may render the MoE framework as a whole less suitable. In this context, our assumption of identifiable clusters in the input distribution is no more restrictive than the foundational assumption of the MoE framework itself. Our model is specifically designed to address scenarios where clusters are challenging to identify due to their varying dependencies on subsets of features. Furthermore, as demonstrated by our additional results involving cluster weight mixing and SoftMoE, our approach continues to perform effectively in settings where expert clusters are highly overlapping.
>
> [1] Witten & Tibshirani, A framework for feature selection in clustering (Journal of the American Statistical Association, 2010)
>
> [2] Robert Jacobs, Michael I Jordan, Steven J Nowlan, and Geoffrey E Hinton. Adaptive mixtures of local experts (Neural computation 1991)
>
> [3] David Eigen, Marc'Aurelio Ranzato, Ilya Sutskever. Learning Factored Representations in a Deep Mixture of Experts (ICLR 2014)

---

> ### Author Response · Authors · 2024-11-18
> **Rebuttal (3 / 3) : Weaknesses 2,3 and Question 2**
>
> **[Weakness 2. The AC router relies on prior layers for initial expert assignments, requiring consistent embedding sizes across layers.]**
>
> **Answer**
>
> In the majority of contemporary deep MoE architectures, the embedding size is typically constant throughout the entire model, and examples of changing embedding sizes are more the exception than the rule. For example, in Switch [4], GLaM [5], SoftMoE [6], and VMoE [7], the embedding size remains the same throughout. Swin is, so far, the only transformer-MoE model we have encountered that features a changing embedding size, but still maintains the same embedding size for 18 of its 22 total layers, so there remains ample opportunity for applying ACMoE in Swin [8].
>
> We note further that many standard-practice architectural features require the same assumption of constant embedding size, such as the residual network. So in this sense, our requirement of constant embedding size at adjacent MoE layers is no more restrictive than what is required for commonplace design choices.
>
> [4] Fedus et al. Switch Transformers: Scaling to Trillion Parameter Models with Simple and Efficient Sparsity (JMLR 2022)
> [5] Du et al. GLaM: Efficient Scaling of Language Models with Mixture-of-Experts (ICML 2022)
> [6] Puigcerver et al. From Sparse to Soft Mixtures of Experts (ICLR 2023)
> [7] Riquelme et al. Scaling Vision with Sparse Mixture of Experts (NeurIPS 2021)
> [8] Liu et al. Swin Transformer: Hierarchical Vision Transformer using Shifted Windows (ICCV 2021)
>
>
> **[Weakness 3. Although tested on large-scale datasets, application to dynamic datasets could strengthen claims of robustness and adaptability.]**
>
> **Answer**
>
> Thanks for your suggestion. We are currently working on validating our method for continual learning tasks to evaluate the ability of our proposed AC router to adapt to dynamically evolving clusters. We will update you with the additional results as soon as have them.
>
> **[Question 2. What mechanisms could be integrated to allow routing without relying on previous layer assignments?]**
>
> **Answer**
>
> We agree that this is an interesting future direction, and we are actively working on this. We do not yet have a finished idea, but we would be happy to share a couple ideas that we think could be helpful for future researchers interested in joining us on this direction.
>
> a) One could estimate the clustering structure of the input tokens without reliance on previous layer expert assignments by applying a clustering algorithm to the tokens before sending them into the router. A straightforward choice would therefore be to use k-means clustering on the input tokens. To reduce the expense of this approach, one could try just one or two steps of k-means.
>
> b) In vision-only settings, one could also try applying an image segmenter before the router to segment the tokens into semantically similar groupings.
>
> We do think it worth noting, however, that we do not see reliance on previous layers as an overly burdensome or problematic scheme for obtaining the cluster assignments. Indeed, this method is efficient, simple to implement, works well empirically, and is justified by the mild assumption that previous layer assignments are good estimates of token cluster membership, especially in later layers of the model. When compared with the alternative ideas discussed above for obtaining the estimates of the cluster membership of the tokens (for example the two above ideas of k-means and image segmenters), we note that our proposed method would be much more efficient.

---

> > ### Author Response · Authors · 2024-11-22
> > **Follow-up on weakness 3**
> >
> > As we work towards setting up experiments in continual learning, could we just verify with the reviewer that this task captures the intention of the question - namely, modeling a dynamic dataset? If not, then if the reviewer could clarify what is meant by a dynamic dataset then that would be much appreciated.
> >
> > Thanks, and hope to hear from you soon regarding above and the rest of our rebuttal. We’d be happy to address any remaining concerns for the remainder of the discussion period.

---

> ### Author Response · Authors · 2024-11-21
> **Any Questions from Reviewer UqYP on Our Rebuttal?**
>
> We would like to thank the reviewer again for your thoughtful reviews and valuable feedback.
>
> We would appreciate it if you could let us know if our responses have addressed your concerns and whether you still have any other questions about our rebuttal.
>
> We would be happy to do any follow-up discussion or address any additional comments.

---

> ### Author Response · Authors · 2024-11-25
> **Additional Results on Dynamic Mixture of Experts in Response to Weakness 3**
>
> We have further tested our AC routing framework in dynamic MoE settings.
>
> Following the approach of [1], we implement ACMoE into top-$p$ dynamic gating. In this setting, rather than routing each token to its top-k highest affinity experts in each MoE layer, we route each token to all experts that have affinity over a certain threshold $p$. This setting permits dynamically activating varying numbers of experts for different tokens at different layers throughout the model. We integrate our AC routing directly into this setting using the same setup as in Section 3 of our manuscript, where the AC routing transformation is computed based on the estimated cluster membership of each token using the top affinity assignment of the previous layer. We present the results for Switch transformer on WikiText-103 language modeling in the following Table A. The same results can be found in Table 20 in Appendix C.10 of our revised manuscript.
>
> **Table A: Results on Top-p Dynamic Routing in Switch Backbone**
>
> | **Model**                        | **Test PPL (↓)** |
> |----------------------------------|------------------|
> | **Fixed top-k routing [2]**          |                  |
> | SMoE-medium (Shazeer et al., 2017) | 35.48           |
> | ACMoE-medium (**Ours**)          | **34.42**        |
> |                                  |                  |
> | **Dynamic top-$p$ routing [1]**        |                  |
> | Switch-Fixed $p$                   | 35.20           |
> | Switch-ACMoE-Fixed $p$ (**Ours**)  | **34.14**        |
> | Switch-Learnable $p$               | 34.29           |
> | Switch-ACMoE-Learnable $p$ (**Ours**) | **33.49**     |
>
> For fixed $p$, we set $p = 0.05$. For learnable $p$, we initialize the parameter to 0.05. We select this initialization as it reproduces approximately similar performance in the Switch backbone under default top-2 routing, thereby aiding direct comparison between fixed top-k and dynamic top-$p$ routing. We see in the dynamic routing setting, ACMoE maintains the same consistent improvement over the Switch baseline of roughly 1 full PPL. These results suggest ACMoE is well-suited to the dynamic MoE setting.
>
> We hope our responses have resolved your concerns. If you believe that our replies have adequately addressed the issues you raised, we kindly ask you to consider whether updating your score would more accurately reflect your updated evaluation of our paper. Thank you once again for your time and thoughtful feedback!
>
> [1] Guo et al. Dynamic mixture of experts: An auto-tuning approach for efficient transformer models (2024)
>
> [2] Shazeer et al. The Sparsely-Gated Mixture-of-Experts Layer (ICLR 2017)

---

### Author Response · Authors · 2024-11-18
**Global Response (1/2)**

Dear AC and reviewers,

Thanks for your thoughtful reviews and valuable comments, which have helped us improve the paper significantly. We are encouraged by the endorsements that: 1) our proposed framework and method are unique and novel (reviewers UqYP, LDye), 2) our theoretical analysis is rigorous (reviewers UqYP, LDye, D6YM), 3) our experimental evaluation is comprehensive (all reviewers), and 4) our method's efficiency makes it easily scalable and integrable into any MoE architecture (reviewers UqYP, LDye).

One common question emerging from reviewers was regarding practical limitations of our method given it requires the embedding size to remain constant across adjacent MoE layers.  Another shared question was on the adaptibility of our method to situations in which latent clusters are not well-defined or the estimated cluster assignments coming from previous layers are noisy. We address these questions here.

**Common embedding size design choices in contemporary MoE architectures.** In the majority of contemporary deep MoE architectures, the embedding size is typically constant throughout the entire model, and examples of changing embedding sizes are more the exception than the rule. For example, in Switch [3], GLaM [4], SoftMoE [1], and VMoE [5], the embedding size remains the same throughout. Swin [2] is, so far, the only transformer-MoE model we've encountered that features a changing embedding size, but still maintains the same embedding size for 18 of its 22 total layers, and so there remains ample opportunity for applying ACMoE.

We note further that many standard-practice architectural features require the same assumption of constant embedding size, such as the residual connection. So in this sense, our requirement of constant embedding size at adjacent MoE layers is no more restrictive than what is required for commonplace design choices.

**ACMoE's adaptibility to settings with overlapping clusters and/or noisy estimated cluster assignments.**
The setting of unreliable or noisy estimated cluster assignments coming from previous layers, and the setting of overlapping clusters can be handled through a straightforward extension of our framework in which we mix the cluster-wise feature weights with mixing proportions corresponding to the affinities in the routing. For example, in a top-2 setting, if $\boldsymbol h$ has affinity scores $\alpha$ and $1-\alpha$ to clusters $k$ and $k'$ respectively, then we could also obtain the required AC routing transformation (Definition 1 in Section 3.1) for $\boldsymbol h$ as $\boldsymbol M_{k^*} = \alpha \boldsymbol M_{k} + (1-\alpha)\boldsymbol M_{k'}$. This approach therefore factors in the confidence with which we believe $\boldsymbol h$ belongs to cluster $k$ or $k'$, and so can be used in settings where we are less sure about the cluster assignment of $\boldsymbol h$ and would prefer to factor in this uncertainty. Furthermore, this adaptation of ACMoE naturally accomodates higher expert granularity backbones (i.e higher top-k settings) or SoftMoE, where all experts are active for every token. In this setting, we have many, overlapping expert clusters and we prefer our routing transformation to model the fact that each token may originate from numerous clusters.

---

> ### Author Response · Authors · 2024-11-18
> **Global Response (2/2)**
>
> To this end, we present in the following tables results for ACMoE integrated into the SoftMoE backbone and further results on the Switch, GLaM, and Swin backbones when applying top-2 cluster mixing. Additionally, we present results for Swin at the larger 0.5B param 'Base' configuration, to offer further evidence of our ACMoE's ability to scale. Results are additionally found in Appendix C.6, C.7, C.8.
>
> **Table 1**: Test Accuracy on ImageNet corrupted PGD, FGSM, and SPSA using SoftMoE [1] backbone
>
> | Model                 | Clean Data |      | PGD      |      | FGSM     |      | SPSA     |      |
> |-----------------------|------------|------|----------|------|----------|------|----------|------|
> |                       | Top 1      | Top 5| Top 1    | Top 5| Top 1    | Top 5| Top 1    | Top 5|
> | *SoftMoE* [1]         | 72.86      | 90.92| 45.29    | 78.91| 56.95    | 85.60| 66.59    | 88.70|
> | Soft-ACMoE (**Ours**) | **73.21**  | **91.23** | **48.25** | **80.49** | **59.01** | **86.69** | **70.63** | **93.22** |
>
> To accomodate the SoftMoE setting, in which all experts are active for each token, we apply cluster weight mixing in ACMoE over the top 8 highest affinity expert clusters. Furthermore, we apply ACMoE on every possible layer of SoftMoE. We see in Table 1 strong, consistent performance gains over SoftMoE, in particular with large robust gains in the range of 6-7%.
>
> **Table 2**: Test Accuracy on ImageNet corrupted PGD, FGSM, and SPSA using Swin Base [2] backbone and cluster mixing over top-2 highest affinity experts
>
> | Model                   | Clean Data |      | PGD      |      | FGSM     |      | SPSA     |      |
> |-------------------------|------------|------|----------|------|----------|------|----------|------|
> |                         | Top 1      | Top 5| Top 1    | Top 5| Top 1    | Top 5| Top 1    | Top 5|
> | *Swin-Base* [2]         | 79.06      | 94.37| 44.61    | 79.20| 59.91    | 87.72| 68.94    | 89.00|
> | Swin-ACMoE-Mix 2-Base (**Ours**) | **79.25**  | **94.42** | **46.28** | **80.24** | **61.78** | **87.55** | **70.28** | **89.38** |
>
> We see in Table 2 that weight mixing in the Swin backbone at the base configuration maintains consistent gains in clean and contaminated performance.
>
> **Table 3**: ACMoE with cluster weight mixing in Switch [3] and GLaM [4]
>
> | Clusters Mixed          | Test PPL (↓) |
> |-------------------------|--------------|
> | *Switch Transformer* [3]| 35.48        |
> | Switch-ACMoE-Mix 2      | 34.66        |
> | Switch-ACMoE-Mix 1  **(Ours)**       | **34.42**    |
> |                         |              |
> | *GLaM* [4]              | 38.27        |
> | GLaM-ACMoE-Mix 2        | **35.29**    |
> | GLaM-ACMoE-Mix 1  **(Ours)**       | 36.26        |
>
> Here, in Table 3, Mix 1 refers to the original result presented in the main text, where we take the top affinity expert cluster as the estimated cluster membership of a given token. Mix 2 refers to ACMoE when applying cluster weight mixing between the top 2 highest affinity experts for each token. For Switch, results are fairly similar across whether we mix or not. Interestingly, however, GLaM-ACMoE improves by almost an entire PPL, which may indicate that in the GLaM architecture, experts learn much broader specializations (i.e expert clusters overlap), and so token-expert matching is best performed in a space transformed according to the specialization of the top 2 highest affinity expert clusters for each token.
>
> In general, results agree with our expectation that cluster mixing, seen both via our additional results on Swin, Switch, and GLaM, and our results on SoftMoE, offers a straightforward method to model the uncertainty in cluster assignments and the setting in which clusters are overlapping. Through this extension, our method can straightforwardly handle both of these settings well and continue to deliver substantive performance gains.
>
> We hope that our rebuttal has helped to clear concerns about our work. We are glad to answer any further questions you have on our submission and we would appreciate it if we could get your further feedback at your earliest convenience.
>
> [1]: Puigcerver et al. From Sparse to Soft Mixtures of Experts (ICLR 2023)
>
> [2]: Liu et al. Swin Transformer: Hierarchical Vision Transformer using Shifted Windows (ICCV 2021)
>
> [3]: Fedus et al. Switch Transformers: Scaling to Trillion Parameter Models with Simple and Efficient Sparsity (JMLR 2022)
>
> [4]: Du et al. GLaM: Efficient Scaling of Language Models with Mixture-of-Experts (ICML 2022)
>
> [5] Riquelme et al. Scaling Vision with Sparse Mixture of Experts (NeurIPS 2021)

---

### Author Response · Authors · 2024-11-18
**Summary of Revisions**

Incorporating comments and suggestions from reviewers, as well as some further empirical studies we believe informative, we summarize here the main changes in the revised paper:

1. We have conducted additional experiments on ACMoE with cluster weight mixing (Appendix C.6). We show in Tables 15 and 16 of Appendix C.6 the results of weight mixing over the top-2 highest affinity experts in Switch and GLaM backbones, where we see similar performance in Switch and a large improvement in GLaM. This straightforward extension of our framework can be used to factor in the confidence with which we believe a token belongs to an expert cluster, and therefore is useful if we believe previous layer expert assignments are noisy. Furthermore, this setup is useful for integrating ACMoE into higher granularity backbones (where we wish to activate a larger number of experts per token), such as SoftMoE, which we discuss in the next point.
2. We have conducted additional experiments on ACMoE in the SoftMoE backbone (Appendix C.7). We present in Tables 17 and 18 of Appendix C.7 performance gains of ACMoE over SoftMoE on clean, adversarially attacked, and out-of-distribution ImageNet-1K, where ACMoE delivers substantial robust performance improvements in the range of 6-7%.
3. We have conducted additional experiments on Swin-ACMoE in a larger 0.5B parameter 'Base' configuration (Appendix C.8). Table 19 in Appendix C.8 shows ACMoE continues to deliver consistent gains in the larger configuration, with robust performance improvements in the range of 3%.
4. We have conducted an empirical assessment of the routing stability (proportion of tokens for which the expert assignments change as the tokens pass through the model) of SMoE, XMoE, StableMoE, and ACMoE on large-scale language modeling in the Switch Transformer backbone. The assessment and details can be found in Figure 5 in Appendix C.9. We see that for a trained model, ACMoE is substantially better at maintaining consistent routing through the model.
5. We have conducted additional baseline experiments using StableMoE [1] at the Switch and GLaM medium configuration in order to add further empirical support for the proposed benefits of our AC routing scheme. We add these results to Table 3 on page 8.
6. We have conducted an additional ablation study in Tables 13 and 14 in Appendix C.5.3 where we replace the diagonal elements of the AC routing transformation with mean 1 normal random variables. Though unrequested, we nonetheless thought such a study offers useful empirical insight and may be of interest to reviewers, as it shows that the performance gains brought about by ACMoE are not simply the result of noise-induced regularization. Tested in large-scale language modeling and image classification, we see in both Swin and Switch that ACMoE substantially outperforms the random ablation model.
7. We have conducted additional experiments in dynamic mixture of experts [2] where we integrate ACMoE into dynamic top-p routing in the Switch backbone. Results can be found in Table 20 of Appendix C.10.

[1] Dai et al. StableMoE: Stable Routing Strategy for Mixture of Experts (ACL 2022)

[2] Guo et al. Dynamic mixture of experts: An auto-tuning approach for efficient transformer models (2024)

---

### Author Response · Authors · 2024-11-25
**Additional Experimental Results on Dynamic Mixture of Experts**

Based on reviewer UqYP's suggestion, we have further tested our AC routing framework in dynamic MoE settings.

Following the approach of [1], we implement ACMoE into top-$p$ dynamic gating. In this setting, rather than routing each token to its top-k highest affinity experts in each MoE layer, we route each token to all experts that have affinity over a certain threshold $p$. This setting permits dynamically activating varying numbers of experts for different tokens at different layers throughout the model. We integrate our AC routing directly into this setting using the same setup as in Section 3 of our manuscript, where the AC routing transformation is computed based on the estimated cluster membership of each token using the top affinity assignment of the previous layer. We present the results for Switch transformer on WikiText-103 language modeling in the following Table A. The same results can be found in Table 20 in Appendix C.10 of our revised manuscript.

**Table A: Results on Top-p Dynamic Routing in Switch Backbone**

| **Model**                        | **Test PPL (↓)** |
|----------------------------------|------------------|
| **Fixed top-k routing [2]**          |                  |
| SMoE-medium (Shazeer et al., 2017) | 35.48           |
| ACMoE-medium (**Ours**)          | **34.42**        |
|                                  |                  |
| **Dynamic top-$p$ routing [1]**        |                  |
| Switch-Fixed $p$                   | 35.20           |
| Switch-ACMoE-Fixed $p$ (**Ours**)  | **34.14**        |
| Switch-Learnable $p$               | 34.29           |
| Switch-ACMoE-Learnable $p$ (**Ours**) | **33.49**     |

For fixed $p$, we set $p = 0.05$. For learnable $p$, we initialize the parameter to 0.05. We select this initialization as it reproduces approximately similar performance in the Switch backbone under default top-2 routing, thereby aiding direct comparison between fixed top-k and dynamic top-$p$ routing. We see in the dynamic routing setting, ACMoE maintains the same consistent improvement over the Switch baseline of roughly 1 full PPL. These results suggest ACMoE is well-suited to the dynamic MoE setting.

We have correspondingly updated our summary of revisions and the current uploaded manuscript.

[1] Guo et al. Dynamic mixture of experts: An auto-tuning approach for efficient transformer models (2024)

[2] Shazeer et al. The Sparsely-Gated Mixture-of-Experts Layer (ICLR 2017)

---

### Author Response · Authors · 2024-11-27
**Summary of Our Discussion with the Reviewers So Far**

Dear Reviewers, Senior Area Chairs, and Area Chairs,

We would like to summarize the revisions we have made so far, incorporating additional results and improvements based on the reviewers' suggestions:

**- [Reviewers UqYP, LDye, D6YM]** We have conducted additional experiments using ACMoE in the SoftMoE backbone evaluated on ImageNet. We present in Tables 17 and 18 of Appendix C.7 performance gains of ACMoE over SoftMoE on clean, adversarially attacked, and out-of-distribution data, where ACMoE delivers substantial robust performance improvements in the range of 6-7%. Incorporating ACMoE into SoftMoE addressed the question of how adaptable our AC routing framework is to settings where latent clusters are modeled as overlapping or indistinct, and adds further justification to the compatibility of our framework with a wide range of MoE backbones.

**- [Reviewers UqYP, LDye]** We have conducted additional experiments using ACMoE in GlaM, Switch, and Swin backbones using cluster weight mixing, which is a straightforward extension of our framework to handle situations where previous layer cluster assignments may be noisy or unreliable. We present the results in Tables 15 and 16 of Appendix C.6, where we see that ACMoE maintains strong performance or even improves with an additional whole PPL in the case of GLaM.

**- [Reviewer UqYP]** We have conducted additional experiments using ACMoE in the Switch backbone using dynamic top-$p$ routing in order to empirically justify the adaptability of ACMoE to the dynamic MoE setting. We present the results in Table 20 in Appendix C.10, where we see ACMoE maintains the same strong, consistent performance gains over Switch transformer.

**- [Reviewer D6YM]** We have conducted additional baseline experiments using StableMoE in the Switch and GLaM medium configuration in order to add further empirical support for the proposed benefits of our AC routing scheme over baseline methods. Results are found in Table 3 of the main text.

**- [Reviewer D6YM]** We have conducted an empirical assessment of the routing stability (proportion of tokens for which the expert assignments change as the tokens pass through the model) of SMoE, XMoE, StableMoE, and ACMoE in the Switch Transformer backbone. The assessment and details can be found in Figure 5 in Appendix C.9. We see that for a trained model, ACMoE is substantially better at maintaining consistent routing through the model.

**- [Reviewers D6YM, LDye]** We have provided ablation studies on layer-wise placement of ACMoE (Tables 10 & 12, Appendix C.5.2), which show that the AC router is able to improve token-expert matching even at early layers in the network. We also ablate the measure of dispersion used in the AC routing transformation (Tables 9 & 11, Appendix C.5.1), finding that the framework is robust to the selected measure of dispersion, but attains top performance when using MAD as reported.

**- [Reviewer LDye]** We have provided theoretical clarification for the justification and broad applicability of our GMM modeling setup used in our robustness propositions.

**- [Reviewers LDye, D6YM, UqYP]** We have provided additional justification for the widespread and easy incorporability of our method into contemporary MoE architectures, and also further justified the benefits of estimating the tokens' cluster membership by previous layer assignments by appealing to the considerable efficiency advantages of the scheme.

**- [Provided without request]** To further enhance the empirical justification of our method, we additionally provided clean and robust results for ACmoE in the larger 0.5B Swin backbone to demonstrate the ability of our method to scale (Table 19, Appendix C.8). We also conducted a random ablation study to demonstrate the performance gains of ACMoE do not stem from noise-induced regularization (Tables 13 & 14, Appendix C.5.3).

We thank the reviewers for their valuable comments, which have helped us improve our paper both theoretically and empirically. We are happy to address any additional comments or questions during the extended discussion phase.

Best regards,

Authors

---

### Meta-Review · Area_Chair_rcMF · 2024-12-08

**Metareview:**

This paper introduces the Adaptive Clustering (AC) router for Mixture-of-Experts (MoE), providing a clustering-based approach to token-expert routing. The method achieves faster convergence, improved robustness, and better performance without added parameters. The theoretical analysis is rigorous, and extensive experiments on language and vision tasks demonstrate significant gains over baselines. The paper is well-written, technically sound, and makes a valuable contribution to MoE research.

**Additional Comments On Reviewer Discussion:**

All reviewers are positive on the paper.

---

### Decision · Program_Chairs · 2025-01-22

Accept (Poster)